# ConE: Cone Embeddings for Multi-Hop Reasoning over Knowledge Graphs

**Zhanqiu Zhang**[1,2]    **Jie Wang**[1,2] [*]    **Jiajun Chen**[1,2]    **Shuiwang Ji**[3]    **Feng Wu**[1,2]

[1]CAS Key Laboratory of Technology in GIPAS
University of Science and Technology of China
[2]Institute of Artificial Intelligence
Hefei Comprehensive National Science Center
{zqzhang,jjchen}@mail.ustc.edu.cn,{jiewangx,fengwu}@ustc.edu.cn
[3]Texas A&M University
sji@tamu.edu

## Abstract

Query embedding (QE)—which aims to embed entities and first-order logical (FOL) queries in low-dimensional spaces—has shown great power in multi-hop reasoning over knowledge graphs. Recently, embedding entities and queries with geometric shapes becomes a promising direction, as geometric shapes can naturally represent answer sets of queries and logical relationships among them. However, existing geometry-based models have difficulty in modeling queries with negation, which significantly limits their applicability. To address this challenge, we propose a novel query embedding model, namely **Con**e **E**mbeddings (ConE), which is the first geometry-based QE model that can handle all the FOL operations, including conjunction, disjunction, and negation. Specifically, ConE represents entities and queries as Cartesian products of two-dimensional cones, where the intersection and union of cones naturally model the conjunction and disjunction operations. By further noticing that the closure of complement of cones remains cones, we design geometric complement operators in the embedding space for the negation operations. Experiments demonstrate that ConE significantly outperforms existing state-of-the-art methods on benchmark datasets.

## 1   Introduction

Multi-hop reasoning over knowledge graphs (KGs)—which aims to find answer entities of given queries using knowledge from KGs—has attracted great attention from both academia and industry recently [24, 23, 16]. In general, it involves answering first-order logic (FOL) queries over KGs using operators including existential quantification ($\exists$), conjunction ($\wedge$), disjunction ($\vee$), and negation ($\neg$). A popular approach to multi-hop reasoning over KGs is to first transform a FOL query to its corresponding computation graph—where each node represents a set of entities and each edge represents a logical operation—and then traverse the KG according to the computation graph to identify the answer set. However, this approach confronts two major challenges. First, when some links are missing in KGs, it has difficulties in identifying the correct answers. Second, it needs to deal with all the intermediate entities on reasoning paths, which may lead to exponential computation cost.

To address these challenges, researchers have paid increasing attention to the query embedding (QE) technique, which embeds entities and FOL queries in low-dimensional spaces [12, 22, 21, 25]. QE models associate each logical operator in computation graphs with a logical operation in embedding

---

[*]Corresponding author.

spaces. Given a query, QE models generate query embeddings following the corresponding computation graph. Then, they determine whether an entity is a correct answer based on similarities between the query embeddings and entity embeddings.

Among the existing QE models, geometry-based models that embed entities and queries into geometric shapes have shown promising performance [12, 11, 5, 22]. Geometry-based models usually represent entity sets as "regions" (e.g., points and boxes) in Euclidean spaces and then design set operations upon them. For example, Query2Box [22] represents entities as points and queries as boxes. If a point is inside a box, then the corresponding entity is the answer to the query. Compared with non-geometric methods, geometric shapes provide a natural and easily interpretable way to represent sets and logical relationships among them.

However, existing geometry-based models have difficulty in modeling queries with negations, which significantly limits their applicability. For example, GQE [12] and Query2Box [22]—which embed queries to points and boxes, respectively—cannot handle queries with negation, as the complement of a point/box is no longer a point/box. To tackle this problem, Ren & Leskovec [21] propose a probabilistic QE model using Beta distributions. However, it does not have some advantages of geometric models. For example, using Beta distributions, it is unclear how to determine whether an entity is an answer to a query as that in the box case [22]. Therefore, proposing a geometric QE model that can model all the FOL queries is still challenging but promising.

In this paper, we propose a novel geometry-based query embedding model—namely, **Con**e **E**mbeddings (ConE)—which represents entities and queries as Cartesian products of two-dimensional cones. Specifically, if the cones representing entities are subsets of the cones representing queries, then these entities are the answers to the query. To perform multi-hop reasoning in the embedding space, we define the conjunction and disjunction operations that correspond to the intersection and union of cones. Further, by noticing that the closure of complement of cones are still cones, we correspondingly design geometric complement operators in the embedding space for the negation operations. To the best of our knowledge, ConE is the first geometry-based QE model that can handle all the FOL operations, including conjunction, disjunction, and negation. Experiments demonstrate that ConE significantly outperforms existing state-of-the-art methods on benchmark datasets.

## 2 Related Work

Our work is related to answering multi-hop logical queries over KGs and geometric embeddings.

**Answering multi-hop logical queries over KGs.** To answer multi-hop FOL queries, path-based methods [30, 17, 10] start from anchor entities and require traversing the intermediate entities on the path, which leads to exponential computation cost. Embedding-based models are another line of works, which embed FOL queries into low-dimensional spaces. For example, existing works embed queries to geometric shapes [12, 22, 11, 5], probability distributions [21], and complex objects [9, 25]. Our work also embeds queries to geometric shapes. The main difference is that our work can handle all the FOL operations, while existing works cannot.

**Other Geometric embeddings.** Geometric embeddings are popular in recent years. For example, geometric operations including translation [3, 18], rotation [28, 26, 14, 32], and complex geometric operations [1, 33] have been widely used in knowledge graph embeddings. Other geometric embedding methods also manage to use boxes [6], convex cones [15, 29, 19], etc. For example, Lütfü Özçep et al. [19] use axis-aligned cones to embed ontologies expressed in the ALC description logic, and use polars of cones to model negation operators. Recent years have also witnessed the development of embeddings in non-Euclidean geometry, such as Poincaré embeddings [1, 20] and hyperbolic entailment cones [8]. Notably, although there exist works that also use cone embeddings [15, 29, 8, 19], they are not designed for the multi-hop reasoning task and their definition of cones are different from that in our work.

## 3 Preliminaries

In this section, we review the background of query embeddings in Section 3.1 and introduce some basic concepts of two-dimensional cones in Section 3.2.

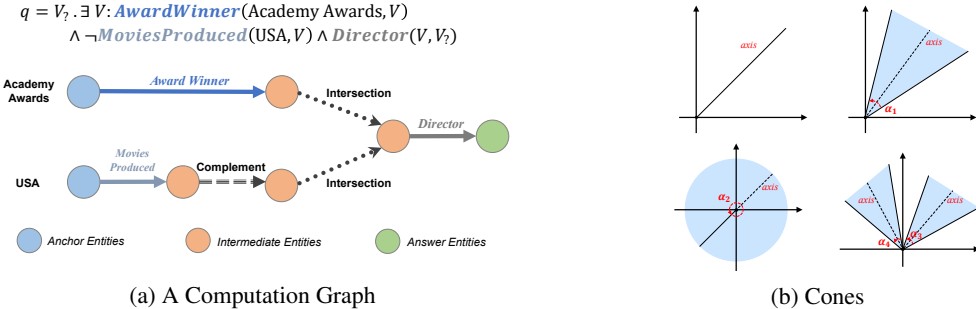

$q = V_? . \exists V : \textbf{\textit{AwardWinner}}(\text{Academy Awards}, V)$
$\wedge \neg \textbf{\textit{MoviesProduced}}(\text{USA}, V) \wedge \textbf{\textit{Director}}(V, V_?)$

(a) A Computation Graph  (b) Cones

Figure 1: Examples of a computation graph and several cones. In Figure 1a, the natural language interpretation of the query is "List all the directors of non-American movies that win the Academy Awards". In Figure 1b, the four cones are: a cone with aperture $0$, a sector-cone with aperture $0 < \alpha_1 < \pi$, a sector-cone with aperture $\alpha_2 = 2\pi$, and a cone that is the union of two sector-cones.

## 3.1 Backgrounds

**Knowledge Graphs (KGs).** Given a set $\mathcal{V}$ of entities (vertices) and a set $\mathcal{E}$ of relations (edges), a knowledge graph $\mathcal{G} = \{(s_i, p_j, o_k)\} \subset \mathcal{V} \times \mathcal{E} \times \mathcal{V}$ is a set of factual triple, where $p_j \in \mathcal{E}$ is a predicate, and $s_i, o_k \in \mathcal{V}$ are subject and object, respectively. Suppose that $r_j(\cdot, \cdot) \in \mathcal{R}$ is a binary function $r_j : \mathcal{V} \times \mathcal{V} \rightarrow \{\text{True}, \text{False}\}$ corresponding to $p_j$, where $r_j(s_i, o_k) = \text{True}$ if and only if $(s_i, p_j, o_k)$ is a factual triples. Then, for all $(s_i, p_j, o_k) \in \mathcal{G}$, we have $r_j(s_i, o_k) = \text{True}$. Note that both $\mathcal{E}$ and $\mathcal{R}$ are involved with relations, while $\mathcal{E}$ is a set of relation instances and $\mathcal{R}$ is a set of relational functions.

**First-Order Logic (FOL).** FOL queries in the query embedding literature involve logical operations including existential quantification ($\exists$), conjunction ($\wedge$), disjunction ($\vee$), and negation ($\neg$). Universal quantification ($\forall$) is not included, as no entity connects with all other entities in real-world KGs [21].

We use FOL queries in its Disjunctive Normal Form (DNF) [7], which represents FOL queries as a disjunction of conjunctions. To formulate FOL queries, we assume that $\mathcal{V}_a \subset \mathcal{V}$ is the non-variable anchor entity set, $V_1, \ldots, V_k$ are existentially quantified bound variables, and $V_?$ is the target variable, i.e., the answers to a certain query. Then, a FOL query $q$ in the disjunctive normal form is:

$$q[V_?] = V_? . \exists V_1, \ldots, V_k : c_1 \vee c_2 \vee \cdots \vee c_n.$$

Specifically, $c_i$ are *conjunctions*, i.e., $c_i = e_{i1} \wedge \cdots \wedge e_{im}$, where $e_{ij} = r(v_a, V)$ or $\neg r(v_a, V)$ or $r(V', V)$ or $\neg r(V', V)$, $v_a \in \mathcal{V}_a, V \in \{V_?, V_1, \ldots, V_k\}, V' \in \{V_1, \ldots, V_k\}$, and $V \neq V'$.

Using the aforementioned notations, answering a query $q$ is equivalent to *finding the set of entities* $[\![q]\!] \subset \mathcal{V}$, where $v \in [\![q]\!]$ if and only if $q[v]$ is True.

**Computation Graphs.** Given a query, we represent the reasoning procedure as a computation graph (see Figure 1a for an example), of which nodes represent entity sets and edges represent logical operations over entity sets. We map edges to logical operators according to the following rules.

- *Relation Traversal→Projection Operator $\mathcal{P}$.* Given a set of entities $S \subset \mathcal{V}$ and a relational function $r \in \mathcal{R}$, the projection operator $\mathcal{P}$ outputs all the adjacent entities $\cup_{v \in S} N(v, r)$, where $N(v, r)$ is the set of entities such that $r(v, v') = \text{True}$ for all $v' \in N(v, r)$.

- *Conjunction→Intersection Operator $\mathcal{I}$.* Given $n$ sets of entities $\{S_1, S_2, \ldots, S_n\}$, the intersection operator $\mathcal{I}$ performs set intersection to obtain $\cap_{i=1}^n S_n$.

- *Disjunction→Union Operator $\mathcal{U}$.* Given $n$ sets of entities $\{S_1, S_2, \ldots, S_n\}$, the union operator $\mathcal{U}$ performs set union to obtain $\cup_{i=1}^n S_n$.

- *Negation→Complement Operator $\mathcal{C}$.* Given an entity set $S \subset \mathcal{V}$, $\mathcal{C}$ gives $\bar{S} = \mathcal{V} \backslash S$.

**Query Embeddings (QE).** QE models generate low-dimensional continuous embeddings for queries and entities, and associate each logical operator for entity sets with an operation in embedding spaces. Since an entity is equivalent to a set with a single element and each query $q$ is corresponding to a unique answer set $[\![q]\!]$, the aim of QE models is equivalent to embedding entity sets that can be answers to some queries.

## 3.2 Cones in Two-Dimensional Spaces

To represent FOL queries as Cartesian products of two-dimensional cones, we introduce some definitions about cones and the parameterization method of a special class of cones.

**Definition 1** (Boyd & Vandenberghe [4]). *A set $C \subset \mathbb{R}^2$ is called a **cone**, if for every $x \in C$ and $\lambda \geq 0$, we have $\lambda x \in C$. A set is a **convex cone** if it is convex and a cone, which means that for any $x_1, x_2 \in C$ and $\lambda_1, \lambda_2 \geq 0$, we have $\lambda_1 x_1 + \lambda_2 x_2 \in C$.*

By letting $\lambda = 0$ in Definition 1, we know that a cone must contain the origin. In view of this property, we define a new operation called *closure-complement* for cones.

**Definition 2.** *Suppose that $C \subset \mathbb{R}^2$ is a cone. Then, the **closure-complement** of $C$ is defined by $\tilde{C} = \boldsymbol{cl}(\mathbb{R}^2 \backslash C)$, where $\boldsymbol{cl}(\cdot)$ is the closure of a set.*

Next, we introduce a class of cones that can be parameterized in a scalable way.

**Definition 3.** *A 2D closed cone is called a **sector-cone**, if its closure-complement or itself is convex.*

The set of sector-cones is closed under closure-complement and their union and intersection are still cones. Besides, we have the following proposition, whose proof is provided in Appendix A.

**Proposition 1.** *A sector-cone is always axially symmetric.*

**Parameterization of 2D Sector-Cones.** Proposition 1 suggests that we can use a pair of parameters to represent a two-dimensional sector-cone:

$$(\theta_{\mathrm{ax}}, \theta_{\mathrm{ap}}), \text{where } \theta_{\mathrm{ax}} \in [-\pi, \pi), \ \theta_{\mathrm{ap}} \in [0, 2\pi].$$

Specifically, $\theta_{\mathrm{ax}}$ represents the angle between the symmetry axis of the sector-cone and the positive $x$ axis. $\theta_{\mathrm{ap}}$ represents the aperture of the sector-cone. For any points in the cone, its phase will be in $[\theta_{\mathrm{ax}} - \theta_{\mathrm{ap}}/2, \theta_{\mathrm{ax}} + \theta_{\mathrm{ap}}/2]$. Figure 1b gives examples of several (sector-)cones. One may notice that sector-cones share some similarities with boxes defined in Query2Box, which also involves region representations. However, we argue that sector-cones are more expressive than boxes, of which the details are provided in Appendix F.

Let $\mathbb{K}$ be the space consisting of all $(\theta_{\mathrm{ax}}, \theta_{\mathrm{ap}})$. We can represent an arbitrary sector-cone $C_0$ as $C_0 = (\theta_{\mathrm{ax}}, \theta_{\mathrm{ap}}) \in \mathbb{K}$. Then, for a $d$-ary Cartesian product of sector-cones

$$C = C_1 \times C_2 \times \ldots \times C_d, \tag{1}$$

we represent it via a $d$-dimensional vector in $\mathbb{K}^d$:

$$C = \left( (\theta_{\mathrm{ax}}^1, \theta_{\mathrm{ap}}^1), \ldots, (\theta_{\mathrm{ax}}^d, \theta_{\mathrm{ap}}^d) \right) \subset \mathbb{K}^d. \tag{2}$$

where $\theta_{\mathrm{ax}}^i \in [-\pi, \pi)$, $\theta_{\mathrm{ap}}^i \in [0, 2\pi]$, for $i = 1, \ldots, d$. Or equivalently, $C = (\boldsymbol{\theta}_{\mathrm{ax}}, \boldsymbol{\theta}_{\mathrm{ap}})$, where $\boldsymbol{\theta}_{\mathrm{ax}} = (\theta_{\mathrm{ax}}^1, \ldots, \theta_{\mathrm{ax}}^d) \in [-\pi, \pi)^d$ and $\boldsymbol{\theta}_{\mathrm{ap}} = (\theta_{\mathrm{ap}}^1, \ldots, \theta_{\mathrm{ap}}^d) \in [0, 2\pi]^d$.

## 4 Cone Embeddings

In this section, we propose **Con**e **E**mbeddings (ConE) for multi-hop reasoning over KGs. We first introduce cone embeddings for conjunctive queries and entities in Section 4.1. Afterwards, we introduce the logical operators and the methods to learn ConE in Sections 4.2 and 4.3.

### 4.1 Cone Embeddings for Conjunctive Queries and Entities

As introduced in Section 3.1, conjunctive queries constitute the basis of all queries in the DNF form. Embeddings of all queries can be generated by applying logical operators to conjunctive queries' embeddings. Thus, we design embeddings for conjunctive queries in this section. We model queries with disjunction using the Union Operator $\mathcal{U}$ in Section 4.2.

In general, the answer entities to a conjunctive query $q$ have similar semantics. For example, answers to the query that "List all the directors of American movies" should all be persons; answers to the query that "List all the Asian cities that ever held Olympic Games" should all be places. If we embed an entity set $[\![q]\!]$ into an embedding space, we expect entities in $[\![q]\!]$ to have similar embeddings. Thus,

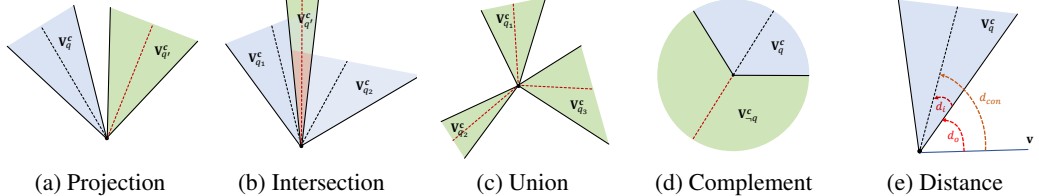

| (a) Projection | (b) Intersection | (c) Union | (d) Complement | (e) Distance |

Figure 2: ConE's logical operators and distance function, of which the embedding dimension $d = 1$.

we expect their embeddings to form a "region" in the embedding space. If the embedding of an entity is inside the region, then the entity is likely to be an answer. Further, we can find a semantic center and a boundary for the region, where the semantic center represents the semantics of $[\![q]\!]$ and the boundary designates how many entities are in $[\![q]\!]$.

To model the embedding region of $[\![q]\!]$, we propose to embed it to a Cartesian product of *sector-cones*. Specifically, we use the parameter $\theta_{\text{ax}}^i$ to represent the semantic center, and the parameter $\theta_{\text{ap}}^i$ to determine the boundary of $[\![q]\!]$. If we use a $d$-ary Cartesian product, i.e., the embedding dimension is $d$, we define the embedding of $[\![q]\!]$ as

$$\mathbf{V}_q^c = (\boldsymbol{\theta}_{\text{ax}}, \boldsymbol{\theta}_{\text{ap}}),$$

where $\boldsymbol{\theta}_{\text{ax}} \in [-\pi, \pi)^d$ are *axes* and $\boldsymbol{\theta}_{\text{ap}} \in [0, 2\pi]^d$ are *apertures*.

An entity $v \in \mathcal{V}$ is equivalent to an entity set with a single element, i.e., $\{v\}$. We propose to represent an entity as a Cartesian product of cones with apertures $0$, where the axes indicates the semantics of the entity. Formally, if the embedding dimension is $d$, the cone embedding of $v$ is $\mathbf{v} = (\boldsymbol{\theta}_{\text{ax}}, \mathbf{0})$, where $\boldsymbol{\theta}_{\text{ax}} \in [-\pi, \pi)^d$ is the axis embedding and $\mathbf{0}$ is a $d$-dimensional vector with all elements being $0$.

## 4.2 Logical Operators for Cone Embeddings

In this section, we introduce our designed logical operators of ConE in the embedding space, including projection, intersection, union, and complement.

It is worth noting that, the composition of logical operators may lead to non-sense queries. For example, the queries "List the intersection/union of American movies and non-American movies" and "List the intersection of American movies and Asian movies" make no sense in real-world applications. However, the main aim of a query embedding model is to represent all entity sets that *can be answer to some real-world query*. Therefore, we do not need to model the entity sets that only correspond to theoretically possible queries [21].

**Projection Operator $\mathcal{P}$.** The goal of $\mathcal{P}$ is to represent an entity's adjacent entities that are linked by a given relation. It maps an entity set to another entity set (see Figure 2a). Thus, we define a relation-dependent function in the embedding space for $\mathcal{P}$:

$$f_r : \mathbb{K}^d \to \mathbb{K}^d, \ \mathbf{V}_q^c \mapsto \mathbf{V}_q^{c'}.$$

We implement $f_r$ in a neural way. First, we represent relations as relational translations of query embeddings and assign each relation with an embedding $\mathbf{r} = (\boldsymbol{\theta}_{\text{ax},r}, \boldsymbol{\theta}_{\text{ap},r})$. Then, we define $f_r$ as

$$f_r(\mathbf{V}_q) = g(\mathbf{MLP}([\boldsymbol{\theta}_{\text{ax}} + \boldsymbol{\theta}_{\text{ax},r}; \boldsymbol{\theta}_{\text{ap}} + \boldsymbol{\theta}_{\text{ap},r}])) \tag{3}$$

where $\mathbf{MLP} : \mathbb{R}^{2d} \to \mathbb{R}^{2d}$ is a multi-layer perceptron network, $[\cdot; \cdot]$ is the concatenation of two vectors, and $g$ is a function that generates $\boldsymbol{\theta}_{\text{ax}}' \in [-\pi, \pi)^d$ and $\boldsymbol{\theta}_{\text{ap}}' \in [0, 2\pi]^d$. We define $g$ as:

$$[g(\mathbf{x})]_i = \begin{cases} \theta_{\text{ax}}'^i = \pi \tanh(\lambda_1 x_i), & \text{if } i \leq d, \\ \theta_{\text{ap}}'^{i-d} = \pi \tanh(\lambda_2 x_i) + \pi, & \text{if } i > d. \end{cases}$$

where $[g(\mathbf{x})]_i$ denotes the $i$-th element of $g(\mathbf{x})$, $\lambda_1$ and $\lambda_2$ are two fixed parameters to control the scale. Note that the range of the hyperbolic tangent function ($\tanh$) are open sets. Thus, we cannot indeed get the boundary value $\theta_{\text{ax}}'^i = -\pi$ and $\theta_{\text{ap}}'^i = 0, 2\pi$. However, when we implement $g$ in experiments, the value of $g$ can be very close to $0$ and $2\pi$, which is equivalent to the closed set numerically.

**Intersection Operator $\mathcal{I}$.** Given a query $q$ that is the conjunction of conjunctive queries $q_i$, the goal of $\mathcal{I}$ is to represent $[\![q]\!] = \cap_{i=1}^n [\![q_i]\!]$. Since the conjunction of conjunctive queries are still conjunctive

queries, the entities in $[\![q]\!]$ should have similar semantics. Recall that we only need to model entity sets that can be answers. We still use a Cartesian product of sector-cones to represent $[\![q]\!]$ (see Figure 2b). Suppose that $\mathbf{V}_q^c = (\boldsymbol{\theta}_{\mathrm{ax}}, \boldsymbol{\theta}_{\mathrm{ap}})$ and $\mathbf{V}_{q_i}^c = (\boldsymbol{\theta}_{i,\mathrm{ax}}, \boldsymbol{\theta}_{i,\mathrm{ap}})$ are cone embeddings for $[\![q]\!]$ and $[\![q_i]\!]$, respectively. We define the intersection operator as follows:

$$\boldsymbol{\theta}_{\mathrm{ax}} = \mathbf{SemanticAverage}(\mathbf{V}_{q_1}^c, \ldots, \mathbf{V}_{q_n}^c),$$
$$\boldsymbol{\theta}_{\mathrm{ap}} = \mathbf{CardMin}(\mathbf{V}_{q_1}^c, \ldots, \mathbf{V}_{q_n}^c),$$

where $\mathbf{SemanticAverage}(\cdot)$ and $\mathbf{CardMin}(\cdot)$ generates semantic centers and apertures, respectively. In the following, we introduce these two functions in detail.

***SemanticAverage.*** As the semantic center of $\mathbf{V}_q^c$, $\boldsymbol{\theta}_{\mathrm{ax}}$ should be close to all the semantic centers $\boldsymbol{\theta}_{i,\mathrm{ax}}$. Thus, we propose to represent $\boldsymbol{\theta}_{\mathrm{ax}}$ as a semantic average of $\boldsymbol{\theta}_{i,\mathrm{ax}}$. We note that the ordinary weighted average may lead to inconsistent semantics. For example, when $d = 1$, if $\boldsymbol{\theta}_{1,\mathrm{ax}} = \pi - \epsilon$ and $\boldsymbol{\theta}_{2,\mathrm{ax}} = -\pi + \epsilon$ ($0 < \epsilon < \pi/4$), then we expect $\boldsymbol{\theta}_{\mathrm{ax}}$ to be around $\pi$. However, if we use the ordinary weighted sum, $\boldsymbol{\theta}_{\mathrm{ax}}$ will be around $0$ with a high probability. To tackle this issue, we propose a semantic average scheme, which takes periodicity of axes into account. For a figure illustration of the difference between the ordinary and semantic average, please refer to Appendix D.

Specifically, we first map $[\boldsymbol{\theta}_{i,\mathrm{ax}}]_j$ to points on the unit circle. Then, compute the weighted average of the points using an attention mechanism. Finally, map the points back to angles that represent axes. Formally, the computation process is

$$[\mathbf{x}; \mathbf{y}] = \sum_{i=1}^{n} [\mathbf{a}_i \circ \cos(\boldsymbol{\theta}_{i,\mathrm{ax}}); \mathbf{a}_i \circ \sin(\boldsymbol{\theta}_{i,\mathrm{ax}})],$$
$$\boldsymbol{\theta}_{\mathrm{ax}} = \mathbf{Arg}(\mathbf{x}, \mathbf{y}),$$

where cos and sin are element-wise cosine and sine functions; $\mathbf{a}_i \in \mathbb{R}^d$ are positive weights vectors that satisfy $\sum_{i=1}^{n} [\mathbf{a}_i]_j = 1$ for all $j = 1, \ldots, d$; $\circ$ is the element-wise multiplication; $\mathbf{Arg}(\cdot)$ is the function that computes arguments of (a Cartesian of) 2D points. Noticing that the weights $\mathbf{a}_i$ are relevant to both axes and apertures, we compute $[\mathbf{a}_i]_j$ via the following attention mechanism:

$$[\mathbf{a}_i]_j = \frac{\exp([\mathbf{MLP}([\boldsymbol{\theta}_{i,\mathrm{ax}} - \boldsymbol{\theta}_{i,\mathrm{ap}}/2; \boldsymbol{\theta}_{i,\mathrm{ax}} + \boldsymbol{\theta}_{i,\mathrm{ap}}/2])]_j)}{\sum_{k=1}^{n} \exp([\mathbf{MLP}([\boldsymbol{\theta}_{k,\mathrm{ax}} - \boldsymbol{\theta}_{k,\mathrm{ap}}/2; \boldsymbol{\theta}_{k,\mathrm{ax}} + \boldsymbol{\theta}_{k,\mathrm{ap}}/2])]_j)},$$

where $\mathbf{MLP} : \mathbb{R}^{2d} \to \mathbb{R}^d$ is a multi-layer perceptron network, $[\cdot; \cdot]$ is the concatenation of two vectors. We can see $\boldsymbol{\theta}_{i,\mathrm{ax}} - \boldsymbol{\theta}_{i,\mathrm{ap}}/2$ and $\boldsymbol{\theta}_{i,\mathrm{ax}} + \boldsymbol{\theta}_{i,\mathrm{ap}}/2$ as the lower and upper bound of sector-cones.

We use $\mathbf{Arg}(\cdot)$ to recover angles of 2D points. Suppose that $\beta_i = \arctan([\mathbf{y}]_i/[\mathbf{x}]_i)$, then

$$[\boldsymbol{\theta}_{\mathrm{ax}}]_i = \begin{cases} \beta_i + \pi, & \text{if } [\mathbf{x}]_i < 0, [\mathbf{y}]_i > 0, \\ \beta_i - \pi, & \text{if } [\mathbf{x}]_i < 0, [\mathbf{y}]_i < 0, \\ \beta_i, & \text{otherwise.} \end{cases}$$

Note that $[\mathbf{x}]_i = 0$ will lead to an illegal division. In experiments, we manually set $[\mathbf{x}]_i$ to be a small number (e.g., $10^{-3}$) when $[\mathbf{x}]_i = 0$.

***CardMin.*** Since $[\![q]\!]$ is the subset of all $[\![q_i]\!]$, $\theta_{\mathrm{ap}}^i$ should be no larger than any apertures $\theta_{j,\mathrm{ap}}^i$. Therefore, we implement $\mathbf{CardMin}$ by a minimum mechanism with cardinality constraints:

$$\theta_{\mathrm{ap}}^i = \min\{\theta_{1,\mathrm{ap}}^i, \ldots, \theta_{n,\mathrm{ap}}^i\} \cdot \sigma([\mathbf{DeepSets}(\{\mathbf{V}_{q_j}\}_{j=1}^n)]_i),$$

where $\sigma(\cdot)$ is the element-wise sigmoid function, $\theta_{j,\mathrm{ap}}^i$ is the $i$-th element of $\boldsymbol{\theta}_{j,\mathrm{ap}}$, $\mathbf{DeepSets}(\cdot)$ is a permutation-invariant function [31]. Specifically, $\mathbf{DeepSets}(\{\mathbf{V}_{q_j}\}_{j=1}^n)$ is computed by

$$\mathbf{MLP}\left(\frac{1}{n}\sum_{j=1}^{n} \mathbf{MLP}\left([\boldsymbol{\theta}_{j,\mathrm{ax}} - \boldsymbol{\theta}_{j,\mathrm{ap}}/2; \boldsymbol{\theta}_{j,\mathrm{ax}} + \boldsymbol{\theta}_{j,\mathrm{ap}}/2]\right)\right).$$

**Union Operator $\mathcal{U}$.** Given a query $q$ that is the disjunction of conjunctive queries $q_i$, the goal of the union operator $\mathcal{U}$ is to represent $[\![q]\!] = \cup_{i=1}^{n}[\![q_i]\!]$. As noted by Ren et al. [22], directly modeling the disjunction leads to unscalable models. Thus, we adopt the DNF technique [22], in which the union operation only appears in the last step in computation graphs.

Suppose that $\mathbf{V}_{q_i}^c = (\boldsymbol{\theta}_{i,\mathrm{ax}}, \boldsymbol{\theta}_{i,\mathrm{ap}})$ are cone embeddings for $[\![q_i]\!]$. To represent the union of several cones (see Figure 2c), we represent $[\![q]\!]$ as a set of $\mathbf{V}_{q_i}^c$:

$$\mathbf{V}_q^d = \{\mathbf{V}_{q_1}^c, \ldots, \mathbf{V}_{q_n}^c\},$$

where $n$ may be various in different queries. Equivalently, $\mathbf{V}_q^d$ can be written as

$$\mathbf{V}_q^d = \left(\{(\theta_{1,\mathrm{ax}}^1, \theta_{1,\mathrm{ap}}^1), \ldots, (\theta_{n,\mathrm{ax}}^1, \theta_{n,\mathrm{ap}}^1)\}, \ldots, \{(\theta_{1,\mathrm{ax}}^d, \theta_{1,\mathrm{ap}}^d), \ldots, (\theta_{n,\mathrm{ax}}^d, \theta_{n,\mathrm{ap}}^d)\}\right).$$

As $\{(\theta_{1,\mathrm{ax}}^i, \theta_{1,\mathrm{ap}}^i), \ldots, (\theta_{n,\mathrm{ax}}^i, \theta_{n,\mathrm{ap}}^i)\}$ are the union of $d$ sector-cones, it is also a cone. Thus, the cone embedding of $q$ is also a Cartesian product of *two-dimensional cones*.

**Complement Operator $\mathcal{C}$.** Given an conjunctive query $q$ and the corresponding entity set $[\![q]\!]$, the aim of $\mathcal{C}$ is to identify the set $[\![\neg q]\!]$, which is the complementary of $[\![q]\!]$, i.e., $\mathcal{V}\backslash[\![q]\!]$. Since the set of sector-cones is closed under closure-complement, we define $\mathcal{C}$ using the closure-complement. Thus, the apertures of $\mathbf{V}_q$ plus the apertures of $\mathbf{V}_{\neg q}$ should be a vector with all elements being $2\pi$. Moreover, to represent the semantic difference between $[\![q]\!]$ and $[\![\neg q]\!]$, we assume that their semantic centers to be opposite. Please refer to Figure 2d for a figure illustration.

Suppose that $\mathbf{V}_q = (\boldsymbol{\theta}_{\mathrm{ax}}, \boldsymbol{\theta}_{\mathrm{ap}})$ and $\mathbf{V}_{\neg q} = (\boldsymbol{\theta}'_{\mathrm{ax}}, \boldsymbol{\theta}'_{\mathrm{ap}})$. We define the complement operator $\mathcal{C}$ as:

$$[\boldsymbol{\theta}'_{\mathrm{ax}}]_i = \begin{cases} [\boldsymbol{\theta}_{\mathrm{ax}}]_i - \pi, & \text{if } [\boldsymbol{\theta}_{\mathrm{ax}}]_i \geq 0, \\ [\boldsymbol{\theta}_{\mathrm{ax}}]_i + \pi, & \text{if } [\boldsymbol{\theta}_{\mathrm{ax}}]_i < 0, \end{cases}$$

$$[\boldsymbol{\theta}'_{\mathrm{ap}}]_i = 2\pi - [\boldsymbol{\theta}_{\mathrm{ap}}]_i.$$

## 4.3 Learning Cone Embeddings

To learn cone embeddings, we expect that the cone embeddings of entities $v \in [\![q]\!]$ are inside the cone embeddings of $q$, and the cone embeddings of entities $v' \notin [\![q]\!]$ are far from the cone embedding of $q$. This motivates us to define a distance function to measure the distance between a given query embedding and an entity embedding, and a training objective with negative sampling.

**Distance Function.** We first define the distance function for conjunctive queries. Inspired by Ren et al. [22], we divide the distance $d$ into two parts—the outside distance $d_o$ and the inside distance $d_i$. Figure 2e gives an illustration of the distance function $d$. Suppose that $\mathbf{v} = (\boldsymbol{\theta}_{\mathrm{ax}}^v, \mathbf{0})$, $\mathbf{V}_q^c = (\boldsymbol{\theta}_{\mathrm{ax}}, \boldsymbol{\theta}_{\mathrm{ap}})$, $\boldsymbol{\theta}_L = \boldsymbol{\theta}_{\mathrm{ax}} - \boldsymbol{\theta}_{\mathrm{ap}}/2$ and $\boldsymbol{\theta}_U = \boldsymbol{\theta}_{\mathrm{ax}} + \boldsymbol{\theta}_{\mathrm{ap}}/2$. We define the distance as

$$d_{con}(\mathbf{v}; \mathbf{V}_q^c) = d_o(\mathbf{v}; \mathbf{V}_q^c) + \lambda d_i(\mathbf{v}; \mathbf{V}_q^c).$$

The outside distance and the inside distance are

$$d_o = \|\min\{|\sin(\boldsymbol{\theta}_{\mathrm{ax}}^v - \boldsymbol{\theta}_L)/2|, |\sin(\boldsymbol{\theta}_{\mathrm{ax}}^v - \boldsymbol{\theta}_U)/2|\}\|_1,$$
$$d_i = \|\min\{|\sin(\boldsymbol{\theta}_{\mathrm{ax}}^v - \boldsymbol{\theta}_{\mathrm{ax}})/2|, |\sin(\boldsymbol{\theta}_{\mathrm{ap}})/2|\}\|_1,$$

where $\|\cdot\|_1$ is the $L_1$ norm, $\sin(\cdot)$ and $\min(\cdot)$ are element-wise sine and minimization functions. Note that as axes and apertures are periodic, we use the sine function to enforce two close angles have small distance. The parameter $\lambda \in (0, 1)$ is fixed during training, so that $\mathbf{v}$ is encouraged to be inside the cones represented by $\mathbf{V}_q^c$, but not necessarily be equal to the semantic center of $\mathbf{V}_q^c$.

Since we represent the disjunctive queries as a set of embeddings, we cannot use $d_{con}$ to directly compute the distance. Nonetheless, the distance between a point and the union of several sets is equal to the minimum distance between the point and each of those sets. Therefore, for a query $q = q_1 \vee \cdots \vee q_n$ in the Disjunctive Normal Form, the distance between $q$ and an entity is

$$d_{dis}(\mathbf{v}; \mathbf{V}_q^d) = \min\{d_{con}(\mathbf{v}; \mathbf{V}_{q_1}^c), \ldots, d_{con}(\mathbf{v}; \mathbf{V}_{q_n}^c)\}.$$

If we use $\mathbf{V}_q$ to represent embeddings of both kinds of queries, the unified distance function $d$ is

$$d(\mathbf{v}; \mathbf{V}_q) = \begin{cases} d_{con}(\mathbf{v}; \mathbf{V}_q), & \text{if } q \text{ is conjunctive queries}, \\ d_{dis}(\mathbf{v}; \mathbf{V}_q), & \text{if } q \text{ is disjunctive queries}. \end{cases}$$

**Training Objective.** Given a training set of queries, we optimize a negative sampling loss

$$L = -\log \sigma(\gamma - d(\mathbf{v}; \mathbf{V}_q)) - \frac{1}{k}\sum_{i=1}^k \log \sigma(d(\mathbf{v}_i'; \mathbf{V}_q) - \gamma),$$

where $\gamma > 0$ is a fixed margin, $v \in [\![q]\!]$ is a positive entity, $v_i' \notin [\![q]\!]$ is the $i$-th negative entity, $k$ is the number of negative entities, and $\sigma(\cdot)$ is the sigmoid function.

Table 1: MRR results for answering queries without negation ($\exists, \wedge, \vee$) on FB15k, FB237, and NELL. The results of BETAE are taken from Ren & Leskovec [21].

| Dataset | Model | 1p | 2p | 3p | 2i | 3i | pi | ip | 2u | up | AVG |
|---------|-------|------|------|------|------|------|------|------|------|------|------|
| FB15k | GQE | 53.9 | 15.5 | 11.1 | 40.2 | 52.4 | 27.5 | 19.4 | 22.3 | 11.7 | 28.2 |
| | Q2B | 70.5 | 23.0 | 15.1 | 61.2 | 71.8 | 41.8 | 28.7 | 37.7 | 19.0 | 40.1 |
| | BETAE | 65.1 | 25.7 | 24.7 | 55.8 | 66.5 | 43.9 | 28.1 | 40.1 | 25.2 | 41.6 |
| | ConE | **73.3** | **33.8** | **29.2** | **64.4** | **73.7** | **50.9** | **35.7** | **55.7** | **31.4** | **49.8** |
| FB237 | GQE | 35.2 | 7.4 | 5.5 | 23.6 | 35.7 | 16.7 | 10.9 | 8.4 | 5.8 | 16.6 |
| | Q2B | 41.3 | 9.9 | 7.2 | 31.1 | 45.4 | 21.9 | 13.3 | 11.9 | 8.1 | 21.1 |
| | BETAE | 39.0 | 10.9 | 10.0 | 28.8 | 42.5 | 22.4 | 12.6 | 12.4 | 9.7 | 20.9 |
| | ConE | **41.8** | **12.8** | **11.0** | **32.6** | **47.3** | **25.5** | **14.0** | **14.5** | **10.8** | **23.4** |
| NELL | GQE | 33.1 | 12.1 | 9.9 | 27.3 | 35.1 | 18.5 | 14.5 | 8.5 | 9.0 | 18.7 |
| | Q2B | 42.7 | 14.5 | 11.7 | 34.7 | 45.8 | 23.2 | 17.4 | 12.0 | 10.7 | 23.6 |
| | BETAE | 53.0 | 13.0 | 11.4 | 37.6 | 47.5 | 24.1 | 14.3 | 12.2 | 8.5 | 24.6 |
| | ConE | **53.1** | **16.1** | **13.9** | **40.0** | **50.8** | **26.3** | **17.5** | **15.3** | **11.3** | **27.2** |

## 5 Experiments

In this section, we conduct experiments to demonstrate that: 1) ConE is a powerful model for the multi-hop reasoning over knowledge graphs; 2) the aperture embeddings of ConE are effective in modeling cardinality (i.e., the number of elements) of answer sets. We first introduce experimental settings in Section 5.1 and then present the experimental results in Sections 5.2 and 5.3. The code of ConE is available on GitHub at `https://github.com/MIRALab-USTC/QE-ConE`.

### 5.1 Experimental Settings

We adopt the commonly used experimental settings for query embeddings [12, 22, 21].

**Datasets and Queries.** We use three datasets: FB15k [2], FB15k-237 (FB237) [27], and NELL995 (NELL) [30]. QE models focus on answering queries involved with incomplete KGs. Thus, we aim to find non-trivial answers to FOL queries that cannot be discovered by traversing KGs. For a fair comparison, we use the same query structures as those in Ren & Leskovec [21]. The training and validation queries consist of five conjunctive structures ($1p/2p/3p/2i/3i$) and five structures with negation ($2in/3in/inp/pni/pin$). We also evaluate models' generalization ability, i.e., answering queries with structures that models have never seen during training. The extra query structures include $ip/pi/2u/up$. Please refer to Appendix B.1 for more details about datasets and query structures.

**Training Protocol.** We use Adam [13] as the optimizer, and use grid search to find the best hyperparameters based on the performance on the validation datasets. For the search range and best hyperparameters, please refer to Appendix B.2.

**Evaluation Protocol.** We use the same evaluation protocol as that in Ren & Leskovec [21]. We first build three KGs: the training KG $\mathcal{G}_{\text{train}}$, the validation KG $\mathcal{G}_{\text{valid}}$, and the test KG $\mathcal{G}_{\text{test}}$ using training edges, training+validation edges, training+validation+test edges, respectively. Given a test (validation) query $q$, we aim to discover non-trivial answers $[\![q]\!]_{\text{test}} \backslash [\![q]\!]_{\text{valid}}$ ($[\![q]\!]_{\text{valid}} \backslash [\![q]\!]_{\text{train}}$). In other words, to answer an entity, we need to impute at least one edge to create an answer path to it. For each non-trivial answer $v$ of a test query $q$, we rank it against non-answer entities $\mathcal{V} \backslash [\![q]\!]_{\text{test}}$. We denote the rank as $r$ and calculate the Mean Reciprocal Rank (MRR), of which the definition is provided in Appendix B.3. Higher MRR indicates better performance.

**Baselines.** We compare ConE against three state-of-the-art models, including GQE [12], Query2Box (Q2B) [22], and BETAE [21]. GQE and Q2B are trained only on five conjunctive structures as they cannot model the queries with negation. Since the best embedding dimension $d$ for ConE is 800, we retrain all the baselines with $d = 800$. The results of GQE and Q2B are better than those reported in Ren & Leskovec [21], while the results of BETAE become slightly worse. Therefore, we reported the results of GQE and Q2B with $d = 1600$ and BETAE with $d = 400$. For the results of BETAE with $d = 800$, please refer to Appendix C.1.

Table 2: MRR results for answering queries with negation on FB15k, FB237, and NELL. The results of BETAE are taken from Ren & Leskovec [21].

| Dataset | Model | 2in | 3in | inp | pin | pni | AVG |
|---------|-------|-----|-----|-----|-----|-----|-----|
| FB15k | BETAE | 14.3 | 14.7 | 11.5 | 6.5 | 12.4 | 11.8 |
|  | ConE | **17.9** | **18.7** | **12.5** | **9.8** | **15.1** | **14.8** |
| FB237 | BETAE | 5.1 | 7.9 | 7.4 | 3.6 | 3.4 | 5.4 |
|  | ConE | **5.4** | **8.6** | **7.8** | **4.0** | **3.6** | **5.9** |
| NELL | BETAE | 5.1 | 7.8 | 10.0 | 3.1 | 3.5 | 5.9 |
|  | ConE | **5.7** | **8.1** | **10.8** | **3.5** | **3.9** | **6.4** |

## 5.2 Main Results

We compare ConE against baseline models on queries with and without negation. We run our model five times with different random seeds and report the average performance. For the error bars of the performance, please refer to Appendix C.5.

**Queries without Negation.** Table 1 shows the experimental results on queries without negation, i.e., existentially positive first-order (EPFO) queries, where **AVG** denotes average performance. Overall, ConE significantly outperforms compared models. ConE achieves on average 19.7%, 12.0%, and 10.6% relative improvement MRR over previous state-of-the-art BETAE on the three datasets, which demonstrates the superiority of geometry-based models. Compared with Q2B, which uses Query2Box to embed queries, ConE gains up to 24.2% relative improvements. ConE also gains an impressive improvement on queries $ip/pi/2u/up$, which are not in the training graph. For example, ConE outperforms BETAE by 38.9% for $2u$ query on FB15k. The results show the superior generality ability of ConE. Since ConE is capable of modeling complement, we can also implement disjunctive queries using De Morgan's law. However, using De Morgan's law always results in sector-cones, which may be inconsistent with the real set union. Thus, the models with DNF outperforms those with De Morgan's law. We include the detailed results in Appendix C.2 due to the space limit.

**Queries with Negation.** Table 2 shows the results of ConE against BETAE on modeling FOL queries with negation. Since GQE and Q2B are not capable of handling the negation operator, we do not include their results in the experiments. Overall, ConE outperforms BETAE by a large margin. Specifically, ConE achieves on average 25.4%, 9.3%, and 8.5% relative improvement MRR over BETAE on FB15k, FB237, and NELL, respectively.

Table 3: Spearman's rank correlation between learned aperture embeddings and the number of queries' answers on FB15k. The results of Q2B and BETAE are taken from Ren & Leskovec [21].

| Model | 1p | 2p | 3p | 2i | 3i | pi | ip | 2in | 3in | inp | pin | pni |
|-------|-----|-----|-----|-----|-----|-----|-----|-----|-----|-----|-----|-----|
| Q2B | 0.30 | 0.22 | 0.26 | 0.33 | 0.27 | 0.30 | 0.14 | - | - | - | - | - |
| BETAE | 0.37 | 0.48 | 0.47 | 0.57 | 0.40 | 0.52 | 0.42 | 0.62 | 0.55 | 0.46 | 0.47 | 0.61 |
| ConE | **0.60** | **0.68** | **0.70** | **0.68** | **0.52** | **0.59** | **0.56** | **0.84** | **0.75** | **0.61** | **0.58** | **0.80** |

## 5.3 Modeling the Cardinality of Answer Sets

As introduced in Section 4.1, the aperture embeddings can designate the cardinality (i.e., the number of elements) of $[\![q]\!]$. In this experiment, we demonstrate that although we do not explicitly enforce ConE to learn cardinality during training, the learned aperture embeddings are effective in modeling the cardinality of answer sets. The property partly accounts for the empirical improvements of ConE.

We compute the correlations between learned aperture embeddings and the cardinality of answer sets. Specifically, for the cone embedding $\mathbf{V}_q = (\boldsymbol{\theta}_{ax}, \boldsymbol{\theta}_{ap})$ of a given query $q$, we use the $L_1$ norm of $\boldsymbol{\theta}_{ap}$ to represent the learned cardinality of $[\![q]\!]$. Then, we compute the Spearman's rank correlation (SRC) between the learned cardinality and the real cardinality, which measures the statistical dependence between the ranking of two variables. Higher correlation indicates that the embeddings can better model the cardinality of answer sets. As we model queries with disjunction using the DNF technique, we do not include the results of disjunctive queries following Ren & Leskovec [21].

Table 3 shows the results of SRC for ConE, Query2Box (Q2B), and BETAE on FB15k. For the results on FB237 and NELL, please refer to Appendix C.3. As Query2Box cannot handle queries with negation, we do not include its results on these queries. On all query structures, ConE outperforms the previous state-of-the-art method BETAE. Note that BETAE is a probabilistic model, of which the authors claim that it can well handle the uncertainty of queries, i.e., the cardinality of answer set. Nonetheless, ConE still outperforms BETAE by a large margin, which demonstrates the expressiveness of cone embeddings. We also conduct experiments using Pearson's correlation, which measures the linear correlation between two variables. Please refer to Appendix C.3 for the results.

## 5.4 The Designed Operators and the Real Set Operations

As introduced in Section 4.2, the designed union (using DNF technique) and complement operators for ConE are non-parametric. They correspond to *exact* set union and complement. Meanwhile, we define neural operators to approximate the projection and intersection operators to achieve a tractable training process and better performance. Notably, the designed neural operators may not exactly match the real set operations. However, experiments on some example cases demonstrate that these neural operators provide good approximations for real set operations. In the following, we show the experimental results for the operators including projection and intersection. In all experiments, the ConE embeddings are trained on FB15k.

**Projection.** Suppose that a set $A$ is included by a set $B$, then we expect the projection of $A$ is also included by the projection of $B$. We randomly generate 8000 pairs of sector-cones $(A_i, B_i)$, where $A_i \subset B_i$. Then, for each $i$, we randomly select a relation $r_i$ and calculate the projections $P_{r_i}(A_i)$ and $P_{r_i}(B_i)$. Ideally, the projected cones should satisfy $P_{r_i}(A_i) \subset P_{r_i}(B_i)$. We calculate the ratio $r_i = |P_{r_i}(A_i) \cap P_{r_i}(B_i)|/|P_{r_i}(A_i)|$ to measure how many elements in $P_{r_i}(A_i)$ are included in $P_{r_i}(A_i) \cap P_{r_i}(B_i)$. Finally, we get an average ratio $r = 0.8113$. That is to say, the learned $P_{r_i}(A_i)$ are included in $P_{r_i}(A_i)$ with a high probability. The learned projection operators approximate the real set projection well.

**Intersection.** To validate that the learned intersection can well approximate real set intersection, we randomly generate 8000 pairs of sector-cones $(C_i, D_i)$, where $C_i \cap D_i$ is not guaranteed to be a sector-cone. Then, we generate embeddings $I(C_i, D_i)$ for the intersection $C_i \cap D_i$. Ideally, the learned cones $I(C_i, D_i)$ should be the same as the real cones $C_i \cap D_i$. We calculate the ratio $r_i = |I(C_i, D_i) \cap (C_i \cap D_i)|/|I(C_i, D_i) \cup (C_i \cap D_i)|$ to measure the overlap between $I(C_i, D_i)$ and $C_i \cap D_i$, and obtain an average ratio of $r = 0.6134$. Note that the experiments are conducted on the test set, and we did not explicitly train our model on these queries. The relatively high overlap ratio $r$ demonstrates that the learned intersection is a good approximation of the real set intersection.

We further conduct experiments to demonstrate that the learned intersection operators can well handle empty intersections. Following Ren et al. [22], on FB15k, we randomly generate $10k$ queries of two types: (a) intersection queries with more than five answers, and (b) intersection queries with empty answer sets. We found that the average aperture is $2.495$ for type (a) queries, while $1.737$ for type (b) queries. The results demonstrate that although we have never trained ConE on the type (b) queries, the empty intersection sets are much more likely to have smaller apertures than queries with non-zero answers (with a $0.9632$ ROC-AUC score). In other words, though we did not train ConE on datasets with empty intersection sets, we can distinguish empty answer sets by the learned apertures.

We also conduct experiments to demonstrate the difference between the learned union operator with De Morgan's law and the real set union. Please refer to Appendix C.6 for details.

## 6 Conclusion

In this paper, we propose a novel query embedding model, namely Cone Embeddings (ConE), to answer multi-hop first-order logical (FOL) queries over knowledge graphs. We represent entity sets as Cartesian products of cones and design corresponding logical operations. To the best of our knowledge, ConE is the first geometric query embedding models that can model all the FOL operations. Experiments demonstrate that ConE significantly outperforms previous state-of-the-art models on benchmark datasets. One future direction is to adapt ConE to queries in the natural language, which will further improve ConE's applicability.

## Acknowledgments and Disclosure of Funding

We would like to thank all the anonymous reviewers for their insightful comments. This work was supported in part by National Science Foundations of China grants 61822604, U19B2026, 61836006, and 62021001, and the Fundamental Research Funds for the Central Universities grant WK3490000004.

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
