# ConE: Cone Embeddings for Multi-Hop Reasoning over Knowledge Graphs

# Appendix

**Zhanqiu Zhang**[1,2]      **Jie Wang**[1,2] *      **Jiajun Chen**[1,2]      **Shuiwang Ji**[3]      **Feng Wu**[1,2]

[1]CAS Key Laboratory of Technology in GIPAS
University of Science and Technology of China
[2]Institute of Artificial Intelligence
Hefei Comprehensive National Science Center
{zqzhang,jjchen}@mail.ustc.edu.cn,{jiewangx,fengwu}@ustc.edu.cn
[3]Texas A&M University
sji@tamu.edu

## A    Proof for Proposition 1

To show Proposition 1, we need the following defition and lemma.

**Definition 1** (Boyd & Vandenberghe [2]). *A cone $C \subset \mathbb{R}^2$ is called a proper cone if it satisfies:*

- *$C$ is convex,*

- *$C$ is closed,*

- *$C$ is solid, which means it has nonempty interior,*

- *$C$ is pointed, which means that it contains no line (or equivalently, $x \in C, -x \in C \Rightarrow x = 0$).*

**Lemma 1** (Seeger & Torki [8]). *Suppose that $K$ is a proper cone. Then*
$$(n-1)^{-1} \leq as(K) \leq 1,$$
*where $as(K)$ is the axial symmetry degree of $K$. The upper bound becomes an equality if and only if $K$ is axially symmetric.*

Be letting $n = 2$ in Lemma 1, we know that $as(K) = 1$, which attains the upper bound. Thus, proper cones in $\mathbb{R}^2$ are always axially symmetric.

**Proposition 1.** *A sector-cone is always axially symmetric.*

*Proof.*  Suppose that $C \subset \mathbb{R}^2$ is a sector-cone, then it is a closed cone.

We further assume that $C$ is convex, contains no line, and has nonempty interior, i.e., it is a proper cone. By Lemma 1, we know that $C$ is axially symmetric. If $C$ is convex but contains a line, i.e., it is the half space, then it is axially symmetric. If $C$ has empty interior, i.e., it is a ray, then it is axially symmetric. Therefore, when $C$ is convex, it is axially symmetric.

If $C$ is not convex, then by the definition of sector-cones, its complement is convex. We know that the closure-complement of $C$, i.e., $\tilde{C}$, is a closed convex cone, and thus axially symmetric. It is easy to see that the axis of symmetry of $\tilde{C}$ is also the axis of symmetry of $C$, and $C$ is axially symmetric.

Therefore, a sector-cone is always axially symmetric.                            $\square$

---

*Corresponding author.

35th Conference on Neural Information Processing Systems (NeurIPS 2021).

Table 1: Statistics of three benchmark datasets, where FB237 denotes FB15k-237, EPFO represents $1p/2p/3p/2i/3i$, and $n1p$ represents $2in/3in/inp/pin/pni$.

| | Training | | Validation | | Test | |
|---|---|---|---|---|---|---|
| **Dataset** | **EPFO** | **Neg** | **1p** | **n1p** | **1p** | **n1p** |
| FB15k | 273,710 | 27,371 | 59,078 | 8,000 | 66,990 | 8,000 |
| FB237 | 149,689 | 14,968 | 20,094 | 5,000 | 22,804 | 5,000 |
| NELL | 107,982 | 10,798 | 16,910 | 4,000 | 17,021 | 4,000 |

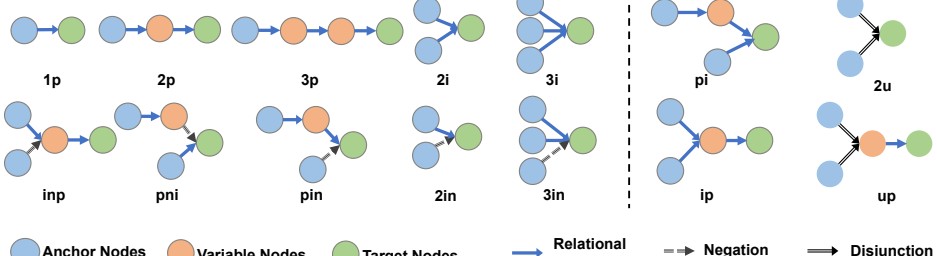

Figure 1: Fourteen queries used in the experiments. where "$p$" denotes relation projection, "$i$" denotes intersection, "$u$" denotes union, and "$n$" denotes negation. The left part of queries is used in the training phase. Both parts are used in the validation and test phases.

# B    More Details about Experiments

In this section, we show more details about experiments that are not included in the main text due to the limited space.

## B.1    Datasets and Query Structures

For a fair comparison, we use the same datasets and query structures as those in Ren & Leskovec [6]. The datasets is created by Ren & Leskovec [6] based on two well-known knowledge graphs Freebase [1] and NELL [3]. They do not contain personally identifiable information or offensive content. Table 1 summarizes the number of different queries in different datasets. Figure 1 shows all the query structures used in the experiments.

## B.2    Training Protocal

We run all the experiments on a single Nvidia Geforce RTX 3090 GPU card. All the models are implemented in Pytorch [5] and based on the official implementation of BETAE [6][2] for a fair comparison. We search the learning rates in $\{5 \times 10^{-5}, 10^{-4}, 5 \times 10^{-4}\}$, the batch size in $\{128, 256, 512\}$, the embedding size in $\{200, 400, 800\}$, the negative sample sizes in $\{32, 64, 128\}$, and the margin $\gamma$ in $\{20, 30, 40, 50, 60\}$. For all the modules using multi-layer perceptron (MLP), we use a three-layer MLP with 1600 hidden neurons and ReLU activation. We apply dropout to the min function in **CardMin** and search the dropout rate in $\{0.05, 0.10, 0.15, 0.20\}$. The best hyperparameters are shown in Table 2.

## B.3    Evaluation Metrics

We choose Mean Reciprocal Rank (MRR) as the evaluation metric. Higher MRR indicates better performance. Definitions are as follows. The mean reciprocal rank is the average of the reciprocal ranks of results for a sample of queries Q:

$$\text{MRR} = \frac{1}{|Q|} \sum_{i=1}^{|Q|} \frac{1}{\text{rank}_i}.$$

---

[2]Link: `https://github.com/snap-stanford/KGReasoning`, licensed under the MIT License.

Table 2: Hyperparameters found by grid search. $d$ is the embedding dimension, $b$ is the batch size, $n$ is the negative sampling size, $\gamma$ is the parameter in the loss function, $m$ is the maximum training step, $l$ is the learning rate, $dr$ is the dropout rate for CardMin, $\lambda_1$ and $\lambda_2$ are weights in the projection operator, $\lambda$ is the parameter in the distance function $d_{con}$.

| | $d$ | $b$ | $n$ | $\gamma$ | $m$ | $l$ | $dr$ | $\lambda_1$ | $\lambda_2$ | $\lambda$ |
|---|---|---|---|---|---|---|---|---|---|---|
| FB15k | 800 | 128 | 512 | $300k$ | 30 | $5 \times 10^{-5}$ | 0.05 | 1.0 | 2.0 | 0.02 |
| FB237 | 800 | 128 | 512 | $300k$ | 30 | $1 \times 10^{-4}$ | 0.10 | 1.0 | 2.0 | 0.02 |
| NELL | 800 | 128 | 512 | $450k$ | 20 | $1 \times 10^{-4}$ | 0.20 | 1.0 | 2.0 | 0.02 |

# C  More Experimental Results

In this section, we give more experimental results that are not included in the main text due to the limited space.

## C.1  Results of BETAE with Embedding Dimension 800

Tables 3 and 4 show the results of BETAE with embedding dimensions 400 (B-400) and 800 (B-800). The results of B-400 is slightly better than that of B-800. Therefore, we report the results of B-400 in the main text.

Table 3: MRR results for answering queries without negation ($\exists$, $\wedge$, $\vee$) on FB15k, FB237, and NELL, where B-400 and B-800 denote BETAE with embedding dimensions 400 and 800, respectively. The results of B-400 models are taken from Ren & Leskovec [6].

| Dataset | Model | 1p | 2p | 3p | 2i | 3i | pi | ip | 2u | up | AVG |
|---|---|---|---|---|---|---|---|---|---|---|---|
| FB15k | B-400 | 65.1 | 25.7 | 24.7 | 55.8 | 66.5 | 43.9 | 28.1 | 40.1 | 25.2 | 41.6 |
| | B-800 | 61.9 | 25.1 | 24.2 | 56.5 | 67.9 | 43.7 | 26.6 | 38.8 | 24.5 | 41.0 |
| | ConE | **73.3** | **33.8** | **29.2** | **64.4** | **73.7** | **50.9** | **35.7** | **55.7** | **31.4** | **49.8** |
| FB237 | B-400 | 39.0 | 10.9 | 10.0 | 28.8 | 42.5 | 22.4 | 12.6 | 12.4 | 9.7 | 20.9 |
| | B-800 | 38.3 | 10.6 | 9.9 | 28.5 | 42.7 | 21.9 | 11.8 | 11.9 | 9.5 | 20.6 |
| | ConE | **41.8** | **12.8** | **11.0** | **32.6** | **47.3** | **25.5** | **14.0** | **14.5** | **10.8** | **23.4** |
| NELL | B-400 | 53.0 | 13.0 | 11.4 | 37.6 | 47.5 | 24.1 | 14.3 | 12.2 | 8.5 | 24.6 |
| | B-800 | 51.6 | 12.5 | 10.7 | 36.9 | 48.2 | 23.3 | 13.9 | 11.8 | 8.1 | 24.1 |
| | ConE | **53.1** | **16.1** | **13.9** | **40.0** | **50.8** | **26.3** | **17.5** | **15.3** | **11.3** | **27.2** |

Table 4: MRR results for answering queries with negation on FB15k, FB237, and NELL, where B-400 and B-800 denote BETAE with embedding dimensions 400 and 800, respectively. The results of B-400 models are taken from Ren & Leskovec [6].

| Dataset | Model | 2in | 3in | inp | pin | pni | AVG |
|---|---|---|---|---|---|---|---|
| FB15k | B-400 | 14.3 | 14.7 | 11.5 | 6.5 | 12.4 | 11.8 |
| | B-800 | 13.7 | 14.6 | 11.3 | 6.4 | 11.9 | 11.6 |
| | ConE | **18.6** | **19.4** | **12.6** | **10.0** | **15.4** | **15.2** |
| FB237 | B-400 | 5.1 | 7.9 | 7.4 | 3.6 | 3.4 | 5.4 |
| | B-800 | 4.9 | 7.1 | 7.6 | 3.7 | 3.3 | 5.3 |
| | ConE | **5.8** | **8.8** | **7.6** | **4.3** | **4.1** | **6.1** |
| NELL | B-400 | 5.1 | 7.8 | 10.0 | 3.1 | 3.5 | 5.9 |
| | B-800 | 5.0 | 7.7 | 10.1 | 3.1 | 2.9 | 5.7 |
| | ConE | **5.6** | **8.1** | **10.9** | **3.5** | **3.9** | **6.4** |

## C.2 Results on Disjunctive Queries

Since ConE is capable of modeling complement, we can also implement disjunctive queries using De Morgan's law, i.e., $\cup_{i=1}^{n} S_i = \overline{\cap_{i=1}^{n} \overline{S_i}}$. Table 5 shows the results of $2u$ and $up$ queries that are implemented using both DNF (**-N**) and De Morgan's law (**-M**). The results show that results of 2u-M/up-M are competitive compared with those of 2u-N/up-N, which all outperform BETAE .

We can also see that ConE using De Morgan's law perform worse than ConE using DNF. The results is reasonable and expectable. If we use the complement to handle queries with unions, their representations will always be sector-cones. However, not all such queries can be well represented by sector-cones (see Figure 3c in the main text).

Table 5: MRR results for answering disjunctive queries on FB15k, FB237, and NELL. The results of BETAE are taken from Ren & Leskovec [6]. **2u-N**/**up-N** indicates that the disjunction is implemented using the DNF technique, while **2u-M**/**up-M** indicates the implementation using De Morgan's law.

| Dataset | Model | 2u-N | 2u-M | up-N | up-M |
|---------|-------|------|------|------|------|
| FB15k | BETAE | 40.1 | 25.0 | 25.2 | 25.4 |
|  | ConE | **55.7** | **37.7** | **31.4** | **29.8** |
| FB237 | BETAE | 12.4 | 11.1 | 9.7 | **9.9** |
|  | ConE | **14.5** | **13.4** | **10.8** | 9.9 |
| NELL | BETAE | 12.2 | 11.0 | 8.5 | 8.6 |
|  | ConE | **15.3** | **14.8** | **11.3** | **10.8** |

## C.3 Correlation Results

Tables 6 and 7 show the results of Spearman's rank correlation between learned embeddings and the number of queries on FB15k-237 and NELL, respectively. The results of Query2Box (Q2B) and BETAE are taken from Ren & Leskovec [6]. The symbol "∗" indicates that the average performance is computed only using results of queries without negation.

Tables 8, 9 and 10 show the results of Pearson correlation between learned embeddings and the number of queries on FB15k, FB15k-237, and NELL, respectively. The results of Query2Box (Q2B) and BETAE are taken from Ren & Leskovec [6]. The symbol "∗" indicates that the average performance is computed only using results of queries without negation.

All the results show that ConE is effective in modeling the cardinality of queries' answer sets.

Table 6: Spearman's rank correlation between learned embeddings and the number of answers on FB15k-237.

| Model | 1p | 2p | 3p | 2i | 3i | pi | ip | 2in | 3in | inp | pin | pni |
|-------|----|----|----|----|----|----|----|-----|-----|-----|-----|-----|
| Q2B | 0.18 | 0.23 | 0.27 | 0.35 | 0.44 | 0.36 | 0.20 | - | - | - | - | - |
| BETAE | 0.406 | 0.50 | 0.57 | 0.60 | 0.52 | 0.54 | 0.44 | 0.69 | 0.58 | 0.51 | 0.47 | 0.67 |
| ConE | **0.70** | **0.71** | **0.74** | **0.82** | **0.72** | **0.70** | **0.62** | **0.90** | **0.83** | **0.66** | **0.57** | **0.88** |

Table 7: Spearman's rank correlation between learned embeddings and the number of answers on NELL.

| Model | 1p | 2p | 3p | 2i | 3i | pi | ip | 2in | 3in | inp | pin | pni |
|-------|----|----|----|----|----|----|----|-----|-----|-----|-----|-----|
| Q2B | 0.15 | 0.29 | 0.31 | 0.38 | 0.41 | 0.36 | 0.35 | - | - | - | - | - |
| BETAE | 0.42 | 0.55 | 0.56 | 0.59 | 0.61 | 0.60 | 0.54 | 0.71 | 0.60 | 0.35 | 0.45 | 0.64 |
| ConE | **0.56** | **0.61** | **0.60** | **0.79** | **0.79** | **0.74** | **0.58** | **0.90** | **0.79** | **0.56** | **0.48** | **0.85** |

Table 8: Pearson correlation between learned embeddings and the number of answers on FB15k.

| Model | 1p | 2p | 3p | 2i | 3i | pi | ip | 2in | 3in | inp | pin | pni |
|-------|-----|-----|-----|-----|-----|-----|-----|-----|-----|-----|-----|-----|
| Q2B | 0.08 | 0.22 | 0.26 | 0.29 | 0.23 | 0.25 | 0.13 | - | - | - | - | - |
| BETAE | 0.22 | 0.36 | 0.38 | 0.39 | 0.30 | 0.31 | 0.31 | 0.44 | 0.41 | 0.34 | 0.36 | 0.44 |
| ConE | **0.33** | **0.53** | **0.59** | **0.5** | **0.45** | **0.37** | **0.42** | **0.65** | **0.55** | **0.50** | **0.52** | **0.64** |

Table 9: Pearson correlation between learned embeddings and the number of answers on FB15k-237.

| Model | 1p | 2p | 3p | 2i | 3i | pi | ip | 2in | 3in | inp | pin | pni |
|-------|-----|-----|-----|-----|-----|-----|-----|-----|-----|-----|-----|-----|
| Q2B | 0.02 | 0.19 | 0.26 | 0.37 | 0.49 | 0.34 | 0.20 | - | - | - | - | - |
| BETAE | 0.23 | 0.37 | 0.45 | 0.36 | 0.31 | 0.32 | 0.33 | 0.46 | 0.41 | 0.39 | 0.36 | 0.48 |
| ConE | **0.40** | **0.52** | **0.61** | **0.67** | **0.69** | **0.47** | **0.49** | **0.71** | **0.66** | **0.53** | **0.47** | **0.72** |

Table 10: Pearson correlation between learned embeddings and the number of answers on NELL.

| Model | 1p | 2p | 3p | 2i | 3i | pi | ip | 2in | 3in | inp | pin | pni |
|-------|-----|-----|-----|-----|-----|-----|-----|-----|-----|-----|-----|-----|
| Q2B | 0.07 | 0.21 | 0.31 | 0.36 | 0.29 | 0.24 | 0.34 | - | - | - | - | - |
| BETAE | 0.24 | 0.40 | 0.43 | 0.40 | 0.39 | 0.40 | **0.40** | 0.52 | 0.51 | 0.26 | **0.35** | 0.46 |
| ConE | **0.48** | **0.45** | **0.49** | **0.72** | **0.68** | **0.52** | 0.39 | **0.74** | **0.66** | **0.38** | 0.34 | **0.69** |

## C.4 Comparison with EmQL

We compare ConE with EmQL [9] on FB15k that is from Query2Box [7]. The dataset is the same as that in EmQL. Table 11 shows that ConE significantly outperforms EmQL and other baselines.

Table 11: Comparison with EmQL [9] under the "generalization" setting. The used dataset FB15k is from Query2Box [7], which is the same as that in EmQL.

|  | GQE | Q2B | BetaE | EmQL | ConE |
|--------|------|------|-------|------|------|
| AVG MRR | 33.2 | 41.0 | 44.6 | 43.9 | **52.9** |

## C.5 Error Bars of Main Results

To evaluate the multi-hop reasoning performance of ConE, we run the model five times with random seeds $\{0, 10, 100, 1000, 10000\}$. In this section, we report the error bars of these results. Table 12 shows the error bar of ConE's MRR results on EPFO queries, i.e., queries without negation. Table 13 shows the error bar of ConE's MRR results on queries with negation. Overall, the standard variances are small, which demonstrate that the performance of ConE is stable.

Table 12: The mean values and standard variances of ConE's MRR results on EPFO queries.

| Dataset | 1p | 2p | 3p | 2i | 3i | pi | ip | 2u | up | AVG |
|---------|------|------|------|------|------|------|------|------|------|------|
| FB | 73.3 ±0.086 | 33.8 ±0.193 | 29.2 ±0.198 | 64.4 ±0.176 | 73.7 ±0.207 | 50.9 ±0.155 | 35.7 ±0.126 | 55.7 ±0.445 | 31.4 ±0.251 | 49.8 ±0.081 |
| FB237 | 41.8 ±0.058 | 12.8 ±0.118 | 11.0 ±0.173 | 32.6 ±0.084 | 47.3 ±0.169 | 25.5 ±0.208 | 14.0 ±0.153 | 14.5 ±0.104 | 10.8 ±0.203 | 23.4 ±0.050 |
| NELL | 53.1 ±0.117 | 16.1 ±0.193 | 13.9 ±0.260 | 40.0 ±0.119 | 50.8 ±0.076 | 26.3 ±0.175 | 17.5 ±0.154 | 15.3 ±0.102 | 11.3 ±0.193 | 27.2 ±0.071 |

Table 13: The mean values and standard variances of ConE's MRR results on queries with negation.

| Dataset | 2in | 3in | inp | pin | pni | AVG |
|---------|-----|-----|-----|-----|-----|-----|
| FB15k | 17.9
±0.158 | 18.7
±0.206 | 12.5
±0.094 | 9.8
±0.428 | 15.1
±0.172 | 14.8
±0.139 |
| FB237 | 5.4
±0.075 | 8.6
±0.076 | 7.8
±0.135 | 4.0
±0.078 | 3.6
±0.069 | 5.9
±0.037 |
| NELL | 5.7
±0.022 | 8.1
±0.129 | 10.8
±0.199 | 3.5
±0.014 | 3.9
±0.088 | 6.4
±0.054 |

### C.6 Union using De Morgan's Law and the Real Union.

When we use De Morgan's law to approximate the union, the resulted cones are always sector-cones, which may be inconsistent with the real union. We conduct experiments to compare the learned embeddings for $\neg(\neg A \wedge \neg B)$ and $A \vee B$. Specifically, we randomly generate 8000 pairs of sector-cones $(A_i, B_i)$ and generate embeddings for $\neg(\neg A_i \cap \neg B_i)$ and $A_i \cup B_i$. Then, to measure the overlap between $\neg(\neg A_i \cap \neg B_i)$ and $A_i \cup B_i$, we calculate the ratio $r_i = |(\neg(\neg A_i \cap \neg B_i)) \cap (A_i \cup B_i)|/|(\neg(\neg A_i \cap \neg B_i)) \cup (A_i \cup B_i)|$, and obtain an average ratio of $r = 0.4618$. The results show a relatively high discrepancy between $\neg(\neg A_i \cap \neg B_i)$ and $A_i \cup B_i$, which again validates the results that ConE with DNF technique can outperform ConE with De Morgan's law.

### C.7 Modeling the Variability of Answer Sets

It is possible that an answer set to a query has a large number of entities but small apertures. When it happens, there are two possible cases.

1. The semantic variability of the entities in this set is low. That is to say, entities in the set closely locate in a cone with a small aperture.

2. Some of the entities are outside the cone. The learned cone embeddings of a query may not include all its answer entities, especially for queries in the validation/test sets. This phenomenon also partly accounts for imperfect performance.

We conduct experiments on FB15k to demonstrate that the learned apertures are correlated with the similarity measures over answer sets. The results are shown in Table 14. In this experiment, suppose that we have an entity set $[[q]] = \{v_1^q, \ldots, v_{n_q}^q\}$ that is the answer set to a query $q$, and its corresponding embeddings $\{\mathbf{v}_1^q, \ldots, \mathbf{v}_{n_q}^q\}$ (note that their apertures are zero). First, we compute the average embeddings of $\{\mathbf{v}_1^q, \ldots, \mathbf{v}_{n_q}^q\}$ using SemanticAverage (all weights are set to be equal) introduced in Section 5.2. Then, we calculate the maximum squared distance from the entity embeddings to the average embeddings and let $\delta_q$ denote the result. That is, $\delta_q$ measures the overall variation of entities in $[[q]]$. Finally, we calculate the Spearman's rank correlation and Pearson's correlation between $\delta_q$ and the learned apertures of $q$.

Table 14: Spearman's rank correlation (SRC) and Pearson's correlation (PC) between learned aperture embeddings and the variety of answer sets on FB15k.

|  | 1p | 2p | 3p | 2i | 3i | pi | ip | 2in | 3in | inp | pin | pni |
|---|-----|-----|-----|-----|-----|-----|-----|-----|-----|-----|-----|-----|
| SRC | 0.28 | 0.27 | 0.22 | 0.58 | 0.43 | 0.44 | 0.27 | 0.41 | 0.42 | 0.10 | 0.16 | 0.39 |
| PC | 0.40 | 0.28 | 0.25 | 0.54 | 0.49 | 0.42 | 0.29 | 0.33 | 0.38 | 0.09 | 0.11 | 0.30 |

## D Semantic Average and Ordinary Average

We give a figure illustration of the difference between the ordinary and semantic average. When $d = 1$, if $\boldsymbol{\theta}_{1,\text{ax}} = \pi - \epsilon$ and $\boldsymbol{\theta}_{2,\text{ax}} = -\pi + \epsilon$ ($0 < \epsilon < \pi/4$), then we expect $\boldsymbol{\theta}_{\text{ax}}$ to be around $\pi$. However, if we use the ordinary weighted sum, $\boldsymbol{\theta}_{\text{ax}}$ will be around 0 with a high probability.

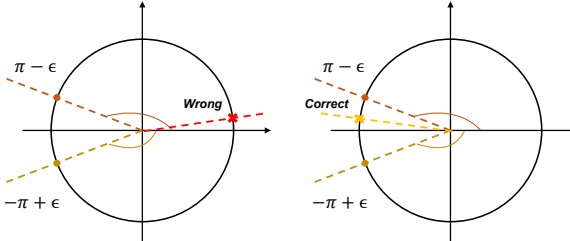

Figure 2: Illustrations of two different weighted average protocols. The left figure represents the ordinary weighted average, which leads to a wrong result. The right figure represents the proposed semantic average.

## E   Determining Whether an Entity Belonging to an Answer Set

Whether an entity $v$ belongs to the answer set $[[q]]$ is determined by the outside distance $d_o(\mathbf{V}, \mathbf{V}_q)$, where $\mathbf{V}$ and $\mathbf{V}_q$ are the cone embeddings for the entity $v$ and query $q$, respectively. Ideally, the entity $v$ belongs to the answer set $[[q]]$ when $d_o(\mathbf{V}, \mathbf{V}_q) = 0$, i.e., the entity embedding $\mathbf{V}$ intersects all of the cones in the query embedding $\mathbf{V}_q$. Accordingly, in the ideal case, an entity belongs to the complement if it intersects all of the cones in the negation query embedding. However, we allow some components of $\mathbf{V}$ outside the corresponding components of $\mathbf{V}_q$ in practice. If $d_o(\mathbf{V}, \mathbf{V}_q)$ is small enough (e.g., smaller than a threshold), we can recognize the entity $v$ as an answer to the query $q$. Moreover, in this way, even if we have two entities that both have mismatched cones, we can say that one entity is more likely to be the answer than the other one by comparing their distances to the query embeddings.

We claim that geometry-based models can determine an entity as an answer to a given query if the cones/boxes represented the entity are inside the cones/boxes represented the query. We conduct experiments to validate the above claim. Specifically, we use trained models ConE/Query2Box with embedding dimensions $d = 800$. That is, each entity and query is represented by a Cartesian product of 800 cones/boxes. Given an entity embedding $\mathbf{v}$ and a query embedding $\mathbf{V}_q$, if a majority (we use a threshold of 500 in the experiments) of the 800 cones/boxes of $\mathbf{v}$ are inside the cones/boxes of $\mathbf{V}_q$, we regard the entity $v$ as an answer to the query $q$. Given a query in the validation/test set, we see its answer entities as positive samples and all the other entities in the KG as negative samples.

Table 15 shows the precision/recall results of the validation/test queries. Note that Query2Box does not apply to queries with negation, so we do not include the corresponding results. The results demonstrate that, using geometry-based models, we can determine whether an entity is an answer to a query by the inclusion relation between entity embeddings and query embeddings. Moreover, ConE outperforms Query2Box on the queries without negation, which is consistent with the results in Table 1 in the main text.

Table 15: Precision/recall results of determining answer entities on queries without negation. The first two rows are results for validation queries, and the last two rows are results for test queries.

|      | FB15k | FB237 | NELL |
|------|-------|-------|------|
| **Q2B**  | 0.490/0.532 | 0.483/0.533 | 0.458/0.581 |
| **ConE** | 0.580/0.678 | 0.519/0.645 | 0.583/0.696 |
| **Q2B**  | 0.502/0.527 | 0.489/0.506 | 0.466/0.562 |
| **ConE** | 0.610/0.670 | 0.536/0.670 | 0.604/0.645 |

Table 16: Precision/recall results of determining answer entities on queries with negation. The first row is results for validation queries, and the last row is results for test queries.

|  | **FB15k** | **FB237** | **NELL** |
|---|---|---|---|
| **ConE** | 0.455/0.648 | 0.545/0.636 | 0.513/0.693 |
|  | 0.510/0.656 | 0.560/0.608 | 0.524/0.627 |

# F   Qualitative Analysis Between ConE and Query2Box

The embedding space and the operators are two key parts of a query embedding model. Therefore, we introduce the superiority of ConE over Query2Box in these two aspects.

## F.1   Embedding Space

Cones can naturally represent a finite universal set and its subset, while Query2Box cannot. The universal set in a knowledge graph corresponds to the set consisting of all the entities, which is finite. As the apertures of cones are bounded (between $0$ and $2\pi$), we can use the cones with apertures $2\pi$ to represent the universal set and find cones with proper apertures to represent any subsets of the universal set. However, since the offsets of boxes in Query2Box are unbounded, how to find boxes to represent the universal set is unclear. It is worth noting that we cannot constrain the offsets of boxes in Query2Box to be bounded, since the composition of its projection operator can generate boxes with arbitrarily large offsets.

The axes of cones are periodic while the centers of boxes are not. It is an important property to model symmetric relations. We will discuss it in detail in the next part.

## F.2   Operators

The operators in query embedding models usually contain projection, intersection, union, and complement. The superiority of ConE over Query2Box mainly comes from the projection and complement operator.

The projection operator of ConE can generate cones with larger or smaller apertures depending on the relation. However, the projection operator of Query2Box always generates a larger box with a translated center, no matter what the relation is. In fact, not all the relation projections should result in larger boxes. For example, if an entity set contains all the countries in the world, and the relation is *contain_cities*, the set of adjacent entities will be larger. If the given entity set contains all cities in the world and the relation is *locate_in_country*, the set of adjacent entities will be smaller. Therefore, the projection operator of ConE is more expressive than that of Query2Box. An expressive projection operator can improve the performance on all the queries as projection appears in all query structures.

The projection operator of ConE can well deal with symmetric relations, while the translation-based projection operator of Query2Box cannot. Suppose that $r$ is a symmetric relation. That is, if $r(h, t)$ is true, then $r(t, h)$ will also be true (e.g., *married_with*). Suppose that the embedding dimension $d = 1$ , the axis of $h$ is $\theta_h$, the axis of $t$ is $\theta_t$, and the apertures of $h$ and $t$ are 0. Then, ConE can model the symmetric relation by learning a neural operator that rotates some axes by an angle $\pi$ and keeps the apertures unchanged. That is, ConE can model the relation $r$ between $h$ and $t$ as $r(\theta_h) = \theta_h + \pi = \theta_t$ and $r(\theta_t) = \theta_t + \pi = \theta_h$, which is benefited from the periodicity. A similar case can be found in RotatE. RotatE can deal with symmetric relations since the phases in complex spaces are periodic.

Since the complements of boxes are no longer boxes, it is still unclear how to use boxes to model the complement operation.

# G   Computational Complexity

The computational complexity of ConE is similar to that of Query2Box [7]. Given a query in Disjunctive Normal Form $q = q_1 \vee \cdots \vee q_n$, where $q_i$ are conjunctive queries, the computational complexity of ConE to answer $q$ is equal to that of answering the $n$ conjunctive queries $q_i$. Answering

$q_i$ requires to execute a sequence of simple geometric cone operations, each of which takes constant time. Then, we perform a fast search using techniques such as Locality Sensitive Hashing [4] to get the final answer.

To evaluate the training speed of ConE and all the baselines, we report the average time spent to run 100 training steps. We run all the models with the same number of embedding parameters using a single RTX 3090 GPU card. Table 17 demonstrates that the simplest model GQE is the most time-efficient. The training speed of ConE is close to that of Query2Box (Q2B) and faster than BetaE.

Table 17: Average time spent per 100 training steps.

| Models | GQE | Q2B | BETAE | ConE |
|---|---|---|---|---|
| **Running Time** | 15s | 17s | 28s | 21s |

## H    Potential Societal Impacts

ConE is a method that performs automatic reasoning over knowledge graphs. One potential negative societal impacts when using automatic reasoning methods (including ConE) is privacy disclosure. If we use public data on the Internet or somewhere else to construct a knowledge graph, and then perform multi-hop reasoning over it, personal information that one does not want to make public may be exposed.