# OpenReview forum: "ConE: Cone Embeddings for Multi-Hop Reasoning over Knowledge Graphs"
_NeurIPS.cc/2021/Conference — NeurIPS 2021 Poster_

### Official Review · Reviewer_jq2v · 2021-07-16

**Rating:** 8
**Confidence:** 5

**Summary:**

This paper represents a novel approach to learning embeddings for entities and relations from a knowledge base that are capable of directly encoding *reasoning* (i.e. FOL). The principal representation proposed for this representation is a *cone* (i.e. a set which is closed under multiplication by a positive scalar) which is motivated by the fact that cones are closed under intersection and complements. In practice, the authors use *sector cones*, which leads to some incongruity between the original motivation and the model in practice (eg. sector cones are not closed under intersection or union). Even so, the authors apply this embedding method to several standard logic embedding tasks and demonstrate significant performance improvements.

**Limitations And Societal Impact:**

The questions I raised above could be viewed as a limitation of the work, particularly if it turns out that the proposed operators are not actually learning the intersection / union / complement operations they claim to represent.

**Main Review:**

### Update after Rebuttal
The authors have performed extensive experiments to aid in understanding how the neural operators approximate their geometric analogs as well as various potential logical discrepancies that exist between the interaction of negation and union / intersection. I appreciate the time and effort which went into these additional experiments, and feel they add significant value to the paper. I would strongly encourage the authors to include this level of detailed analysis in a polished way in the final submission.

### Overview

**Originality:** The approach of using this form of cones is significantly novel, and the approaches to representing relations, intersections, and unions combine well-known techniques in a reasonable way.

**Quality:** The most concerning aspect from the perspective of quality is the distinction between motivation and practice, as mentioned in the summary. Not only is the set of sector cones not closed under intersection or union, the actual operations used for projection, intersection, and union are not necessarily self-consistent (see below). That being said, the empirical evidence in support of the resulting model, regardless of it's departure from the initial motivations, is quite strong.

**Clarity:** The submission is reasonably clear, and gives enough mathematical detail to reproduce the results.

**Significance:** The authors report strong results on the task of logic embeddings, which is of significant interest to the community.

---

### Specific Comments / Concerns

The fundamental question which remains in my mind is whether the original motivation of using *cones* is responsible for the improved performance, particularly when considering the fact that many of the original motivating principles to suggest their use are not applicable to the model which was employed in practice. For example:

1. The authors claim that the aperture of a cone should represent the number of entities it contains, however this aperture must also somehow depend on the variety of answers. For example, consider a query that represents a very specific set of things that (from the KB's perspective) are all very similar vs. a very general query with a wide variety of answers. Here is an approach to test this: pick an arbitrary entity $x$ from the KB, and artificially augment the training data by creating many new entities $\{\tilde x_i\}$ which have identical relations to $x$. Then, any query which contains $x$ suddenly contains many more entities, but this does not actually require that it have a larger aperture - indeed, the exact same model should work if the $\tilde x_i$ entities simply have the same embedding as $x$.
2. As mentioned in the summary, the use of cones is motivated by their closure under intersection and union, however the actual model uses sector cones which are not closed under intersection and union. Still, it seems the model performs very well. The authors mention that they have resubmitted this paper to NeurIPS with additional motivation and evaluation along these lines, however I actually feel it is simply less emphasized in the beginning and has not been fully explored. For example, the representations could be assessed as to how well they approximate the actual intersection and union operator - if it is close, then perhaps the original motivation is justified; if it is not, then the next question on my mind would be to explore the actual model and try to identify which properties are contributing to the increased performance.
3. Apart from the difference between cones and sector cones, the operations themselves are all learned and do not necessarily obey reasonable compositional semantics. For example, if cone A is inside cone B (indicating that it contains elements which are, themselves, a subset of those in B) then the projection of A should be inside the projection of B, but this is not guaranteed by the current model. Composing operations lead to additional issues, for example, the complement of intersections may be far from the union of their complements. This can be assessed in much the same way as proposed above - compare the learned operators with geometrically accurate versions; if it is close, then perhaps it is learning to approximate the geometric idealized model, if it is not close then I would argue that it is not, fundamentally, embedding the logical operations it claims to.

**Time Spent Reviewing:**

3

---

> ### Author Response · Authors · 2021-08-10
> **Response to Reviewer jq2v (1/2)**
>
> We thank the reviewer for the insightful comments. We address the concerns in detail as follows.
>
>
>
> 1. **Motivation**
>    - First, we clarify some misunderstandings about the motivation of ConE.
>
>      1. The motivation of ConE is not that cones are closed under intersection and complements. Instead, as described in Abstract and Introduction, the motivation is that:
>         1. entities and queries can be represented as Cartesian products of two-dimensional cones;
>         2. the conjunction and disjunction operations naturally correspond to the intersection and union of cones;
>         3. by further noticing that the closure of complement of cones remains cones, we design geometric complement operators in the embedding space for the negation operations.
>
>      2. Our ConE model does not only use sector-cones to represent queries. When representing the disjunction of conjunctive queries, the cone embedding is a Cartesian product of two-dimensional cones instead of sector-cones (see Lines 222-224 in our paper).
>
>    - Then, we emphasize the following facts.
>
>      1. The main aim of a query embedding model is to represent all entity sets that can be answers to some real-world query and we do not need to model the entity sets that only correspond to theoretically possible queries (see Lines 167-172 in our paper). Since not all theoretically possible conjunction of conjunctive queries can appear in real-world applications (e.g., "List the intersection of fruits and sports"), a geometric shape does not have to be closed under all possible intersections. Instead, it is enough that the geometric shapes can represent all entity sets that can be answers to some real-world query in the embedding spaces in a tractable way.
>      2. As we use FOL queries in their DNF form, the union operation only appears in the last step of a query. Therefore, whether the set of cones is closed under union does not affect the model's scalability and performance. Moreover, the embedding spaces of many existing query embedding models, including Query2Box (boxes) and BetaE (Beta distributions), are not closed under union.
>
>    - Finally, based on the two parts above, we restate our motivation as follows.
>
>      1. By noticing that conjunctive queries usually have answer entities with similar semantics, we represent *conjunctive queries* with Cartesian products of *sector-cones*. (Section 4.1)
>      2. The disjunction of conjunctive queries naturally corresponds to the union of sector-cones. As the union of sector-cones is still cones, we represent the disjunction of conjunctive queries as Cartesian products of *cones*. (Section 4.2)
>      3. The conjunction of conjunctive queries is still conjunctive, and thus its answer entities have similar semantics. Recall that we only need to represent entity sets that can be answers to real-world queries and entities with similar semantics should locate in a sector-cone. We can assume that the intersection of sector-cones---which represent conjunctive queries---is also a sector-cone. (Section 4.2)
>      4. The property that the closure of complement of cones remains cones gives us a natural solution to represent the negation of conjunctive queries. (Section 4.2)
>
>
>
> 2. **The aperture of a cone and the number of entities it contains**
>
>    - We do not claim that "the aperture of a cone should represent the number of entities it contains". Although we expect the apertures to designate how many entities are in an answer set, we do not assume that larger apertures have to correspond to more entities. As noted in Line 309 in our paper, "we do not explicitly enforce ConE to learn cardinality during training".
>
>    - If $x$ is an answer entity to the query $q$, adding entities $\tilde{x}_i$ that have the same embeddings as $x$ to the knowledge graph will definitely enlarge the number of answer entities. Nonetheless, experiments in Section 5.3 still show a high correlation between the learned aperture embeddings and the number of answer entities. The results demonstrate that not much of such entities $\tilde{x}_i$ appear in the real-world knowledge graphs.
>
>
>
> 3. **How well the designed operators approximate the real set operations**
>
>    - The intersection, union, and complement operators
>      - The designed union and complement operators for ConE are non-parametric. They correspond to exact set union and complement. Please refer to Lines 216-224 and Lines 225-231 in our paper for the detailed definitions of these two operators.
>      - Approximating the intersection operation in a neural way is widely used in query embeddings models, including Query2Box and BetaE. Moreover, as the authors of Query2Box described in their response to the reviewers (see "Re: Official Blind Review #2 (1/2)" in the [OpenReview](https://openreview.net/forum?id=BJgr4kSFDS&noteId=SyxzjxvjsH) page of Query2Box), "richer learnable parameterization of the intersection operator is more expressive and also robust to noise in the knowledge graphs".
>      - We conduct an experiment to evaluate how well the neural operator approximate the exact set intersection. We randomly generate 8000 pairs of sector-cones $(C_i, D_i)$, where $C_i\cap D_i$ is also a sector-cone. Then, we use a ConE model trained on FB15k to generate embeddings $I(C_i, D_i)$ for the intersection $C_i\cap D_i$. Ideally, the learned cones $I(C_i, D_i)$ should be the same as the real cones $C_i\cap D_i$. We calculate the ratio $r_i=|I(C_i, D_i)\cap (C_i\cap D_i)|/|I(C_i, D_i)\cup (C_i\cap D_i)|$ to measure the overlap between $I(C_i, D_i)$ and $C_i\cap D_i$, and obtain an average ratio of $r=0.7436$. That is to say, the learned intersection $I(C_i, D_i)$ is a good approximation of the real intersection $C_i\cap D_i$.
>      - Recall that it is unnecessary to exactly model all the theoretically possible queries, and we only need to model the case that the intersection is still a sector-cone. For the example "the complement of intersection and the union of their complements" in the review, we can assume that the intersection is a sector-cone. As the union and complement operators for ConE correspond to exact set union and complement, the difference between the learned sets and the real sets comes from the intersection operator, which has been shown to be able to well approximate the real set operations.
>
>    - The projection operator
>      - The inclusion relation can be captured by the learning process of ConE. For example, we expect that, if a cone $A$ is inside cone $B$, then the projected cone $P(A)$ should be inside the projected cone $PB$. When we are training ConE, for any entity $e$ in $P(A)$, it is also a positive sample for $P(B)$. Therefore, the entity $e$ will be encouraged to be inside both $P(A)$ and $P(B)$ by optimizing the loss function defined in Line 251. Moreover, if entities in a cone $C$ is a proper subset of a cone $D$, then we expect that in the embedding space, the cone $C$ is properly included in the cone $D$. By the definition of proper subsets, we know that there exists an entity $e’$ in $D$ that is not in $C$. Thus,  $e’$ will be a negative sample for $C$ and it will be encouraged to be outside $C$ but inside $D$ by optimizing the loss function. In a word, the boundaries of cones will be adjusted to satisfy the set relations via the learning process with positive and negative samples. Other composition semantics that appear in real-world applications can also be captured by this kind of learning process.
>      - We also conduct experiments that ConE models relation projection as the axis rotation and argument translation, which ensures that "if cone $A$ is inside $B$ then the projection of A should be inside the projection of B". However, the model performs much worse than that with the neural operator as presented in the current submission (Average MRR: $0.498$ vs. $0.413$ on FB15k).
>      - We conduct experiments to demonstrate that the learned geometric projection operators are close to its geometric accurate versions. The used ConE model is trained on FB15k. We randomly generate 8000 pairs of sector-cones $(A_i, B_i)$, where $A_i\subset B_i$. Then, for each $i$, we randomly select a relation $r_i$ and calculate the projections $P_{r_i}(A_i)$ and $P_{r_i}(B_i)$. Ideally, the projected cones should satisfy $P_{r_i}(A_i)\subset P_{r_i}(B_i)$. We calculate the ratio $r_i=|P_{r_i}(A_i)\cap P_{r_i}(B_i)|/|P_{r_i}(A_i)|$ to measure how many elements in $P_{r_i}(A_i)$ are included in $P_{r_i}(A_i)\cap P_{r_i}(B_i)$. Finally, we get an average ratio $r=0.8113$. That is to say, the learned  $P_{r_i}(A_i)$ are included in $P_{r_i}(A_i)$ with a high probability.

---

> > ### Author Response · Authors · 2021-08-10
> > **Response to Reviewer jq2v (2/2)**
> >
> > 4. **Which properties are contributing to the increased performance**
> >    - **Comparison with Query2Box**
> >      - The embedding space and the operators are two key parts of a query embedding model. Therefore, we introduce the superiority of ConE over Query2Box in these two aspects.
> >
> >      - Embedding Space
> >
> >        - Cones can naturally represent a finite universal set and its subset, while Query2Box cannot. The universal set in a knowledge graph corresponds to the set consisting of all the entities, which is finite. As the apertures of cones are bounded (between $0$ and $2\pi$), we can use the cones with apertures $2\pi$ to represent the universal set and find cones with proper apertures to represent any subsets of the universal set. However, since the offsets of boxes in Query2Box are unbounded, how to find boxes to represent the universal set is unclear. It is worth noting that we cannot constrain the offsets of boxes in Query2Box to be bounded, since the composition of its projection operator can generate boxes with arbitrarily large offsets.
> >        - The axes of cones are periodic while the centers of boxes are not. It is an important property to model symmetric relations. We will discuss it in detail in the next part.
> >
> >      - Operators
> >
> >        - The projection operator of ConE can generate cones with larger or smaller apertures depending on the relation. However, the projection operator of Query2Box always generates a larger box with a translated center, no matter what the relation is.  In fact, not all the relation projections should result in larger boxes. For example, if an entity set contains all the countries in the world, and the relation is *contain_cities*, the set of adjacent entities will be larger. If the given entity set contains all cities in the world and the relation is *locate_in_country*, the set of adjacent entities will be smaller. Therefore, the projection operator of ConE is more expressive than that of Query2Box. An expressive projection operator can improve the performance on all the queries as projection appears in all query structures.
> >        - The projection operator of ConE can well deal with symmetric relations, while the translation-based projection operator of Query2Box cannot. Suppose that $r$ is a symmetric relation. That is, if $r(h,t)$ is true, then $r(t,h)$ will also be true (e.g., *married_with*).  As a corollary, $r^{2n}(h,h)$ should be true for all positive integer $n$.  Using Query2Box, the centers and offsets of the relation $r$ should all be zero. Otherwise, if the offsets are larger than zero, the projected boxes will contain more and more irrelevant entities as $n$ increases. Given that the offsets are zero, the centers of $r$ should also be zero as the centers of $h$ plus $2n$ times centers of $r$ is still the centers of $h$. Therefore, Query2Box has trouble in modeling symmetric relations as it will represent all symmetric relations as zero vectors. For ConE, we can model the symmetric relations by learning a neural operator that rotates some axes by an angle $\pi$ and keeps the apertures unchanged.
> >
> >    - **Comparison with BetaE for negation**
> >      - ConE outperforms BetaE on queries with negation as ConE can represent the universal set in a knowledge graph. By definition, the computation of set complement is closely related to the universal set. Accordingly, to better model queries with negation, a query embedding model should be capable of representing the universal set. For ConE, we can use cones with apertures $2\pi$ to represent the universal set. However, for BetaE, it is unclear how to use a Beta distribution to achieve the goal.

---

> > ### Comment · Reviewer_jq2v · 2021-08-24
> > **Thank you for the additional experiments related to geometric intersection and preservation of containment under projection**
> >
> > Thank you for the detailed reply.
> >
> > Your reply defends the theoretical weaknesses of ConE in two ways:
> >
> > 1. Deficiencies in representation are not observed in "real-world" queries / datasets.
> >
> >     In particular, this is used as a defense for why it is reasonable to represent queries using a sector cone as opposed to a cone (because conjunctions present in real-world queries would have answers which are semantically similar, and thus lie in a sector cone). The queries tested in the Q2B dataset are generated synthetically, however, and many correspond to very contrived questions when converted back to real-world data. Furthermore, while "fruits and sports" seems an unrealistic query, of course one could imagine plenty of real-world queries for which the intersection set is empty (eg. "name all Turning award winners who won an oscar") and it is important the model is able to address this. Perhaps what is actually being observed here is a bias in the form of synthetic query generation, and the ability for your model to exploit it.
> >
> >     This is also used as a defense for why the conclusions regarding cone aperture correlating with number of entities is reasonable. I realize that this was not a strict requirement of the model, however line 331 seems to assert that it is, nevertheless, partly responsible for the model's improved performance. My point here is simply that (just as with BetaE and Query2Box) this cannot be the whole story, that the aperture must somehow also capture the semantic variability in the answer sets and not just the number of entities - consider the set used cars vs. new cars at a car dealership, where the total number might be similar but the used cars will contain many more makes and models. My example here (artificially duplicating a particular node) was simply intended to expose an extreme case of this, not to claim that this specific case needs to be addressed. To be clear, I don't feel the paper's conclusion is wrong (as number of entities clearly correlates highly with semantic variation, which itself is harder to mathematically define) however this is an area worth additional analysis. For example:
> >     - Qualitatively inspect the sets which violate the claim (eg. high number of entities but low aperture, or the opposite)
> >     - Aggregate measures of similarity (eg. the entity embeddings in this model, or one defined synthetically based on the graph such as the proportion of shared queries) over answer sets, and calculate correlation of these measures with the apertures
> >
> > 2. Training will ensure consistency wherever the model itself does not.
> >
> >     This is used as a defense for why the neural intersection operator is reasonable, and why the fact that the projection operator does not preserve cone containment is a problem. I am quite appreciative of the additional experiments along these lines, which compare the resulting neural operation to the geometric one, and feel these would make a very valuable addition to the paper. I would strongly suggest that additional (even adversarial) experiments along these lines be conducted - for example, in the paper as well as your rebuttals you have claimed that the ConE model has additional representational capacity in the form of it's ability to model negations, however this representation may be inconsistent (and, I would guess, significantly so) from one obtained via disjunction (i.e. comparing $\neg(A \wedge B)$ vs. $\neg A \vee \neg B$).
> >
> > Your claim that "We can assume that the intersection of sector-cones---which represent conjunctive queries---is also a sector-cone." bridges both the above points - namely, that for real-world queries their answers will be semantically similar and thus training will ensure that the cone representations are such that their geometric intersection is also a sector cone. You then mention in your experiment of how well the neural intersection approximates the real one you selected cones for which their intersection was a sector cone, and so of course I am curious as to how often this was not the case. If the number is significant then presumably the claim does not hold for the resulting model. Even if it is not significant, this also biases the aforementioned experiment, since you are avoiding the particularly challenging cases. (Note that it should still be possible to calculate the difference from the actual intersection by using inclusion / exclusion on a union of cones.)
> >
> > I agree that a "softer" model which does not adhere exactly to geometric constraints may be easier to train as well as providing a richer representation, however I think these departures should be acknowledged "up-front" and subsequently analyzed to verify the model is behaving in alignment with geometric motivation. Similarly, any points of structural inconsistency (eg. lack of transitivity as a result of the projection operator) should be harshly examined empirically to ensure that what the model *actually* learns is relatively consistent. I would strongly encourage you to consider including additional experiments along these lines (some specific options provided above).

---

> > > ### Author Response · Authors · 2021-08-26
> > > **Thanks for your further comments**
> > >
> > > We appreciate your time in reading our response and further comments. Below is our response to the comments. We hope it can properly address your concerns.
> > >
> > > 1. **Modeling real-world queries for which the intersection set is empty**
> > >
> > >    - When the intersection set is empty, we expect the corresponding learned cone embeddings to have zero apertures. As shown in Figure 1b in our paper, a cone with a zero aperture (i.e., a ray from the origin) is also a sector-cone. Thus, representing empty sets as cones with zero apertures is consistent with our modeling that represents conjunctive queries as sector-cones.
> > >    - One may argue that we also represent entities as cones with zero apertures, and thus the cones with zero apertures correspond to a single entity instead of empty sets. However, since the set of entities in a knowledge graph is finite, not all possible embeddings correspond to an entity. The possibility that the axes of a given entity are the same as the axes of the empty set is zero. Thus, the modeling mentioned above is reasonable. Moreover, this phenomenon also happens in previous models. For example, Query2Box represents empty sets as boxes with offsets zero, which are also used to represent entities.
> > >    - Similar to Query2Box (please refer to the bottom part of [“Re: Official Blind Review #2 (1/2) "](https://openreview.net/forum?id=BJgr4kSFDS&noteId=SyxzjxvjsH) on their OpenReview page), we do not explicitly ensure that the intersection of non-overlapping cones has zero apertures. Instead, if we can explicitly train ConE on intersection queries whose answer sets are empty, we expect the learned apertures to be almost zero by the DeepSets operation in the neural intersection operator.
> > >    - We also conduct experiments to demonstrate that, although we did not train ConE on datasets with empty intersection sets, we can distinguish empty answer sets by the learned apertures. Following [Query2Box](https://openreview.net/forum?id=BJgr4kSFDS&noteId=SyxzjxvjsH), on FB15k, we randomly generate $10k$​ queries of two types: (a) intersection queries with more than five answers, and (b) intersection queries with empty answer sets. We found that the average aperture is $2.495$​ for type (a) queries, while $1.737$​ for type (b) queries. The results demonstrate that although we did not train ConE on the type (b) queries, the empty intersection sets are much more likely to have smaller apertures sizes than queries with non-zero answers (with a $0.9632$​ ROC-AUC score).
> > >
> > >
> > >
> > > 2. **Additional analysis that the apertures somehow also capture the semantic variability in the answer sets and not just the number of entities**
> > >
> > >    - When an answer set to a query has a large number of entities but small apertures, there are two possible cases.
> > >
> > >      1. The semantic variability of the entities in this set is low. That is to say, entities in the set closely locate in a cone with a small aperture.
> > >      2. Some of the entities are outside the cone. The learned cone embeddings of a query may not include all its answer entities, especially for queries in the validation/test sets. This phenomenon also partly accounts for imperfect performance.
> > >
> > >    - We conduct experiments on FB15k to demonstrate that the learned apertures are correlated with the similarity measures over answer sets. The results are shown in the following table. In this experiment, suppose that we have an entity set $[[q]]=\\{v_1^q, \dots, v_{n_q}^q\\}$​​​​​​ that is the answer set to a query $q$​​​​​​,  and its corresponding embeddings $\\{\textbf{v}\_1^q,\dots,\textbf{v}_{n_q}^q\\}$​​​​​​ (note that their apertures are zero). First, we compute the average embeddings of $\\{\textbf{v}\_1^q,\dots,\textbf{v}\_{n_q}^q\\}$​​​​​​ using SemanticAverage (all weights are set to be equal) introduced in Section 5.2. Then, we calculate the maximum squared distance from the entity embeddings to the average embeddings and let $\delta_q$​​​​​​ denote the result. That is, $\delta_q$​​​​​​ measures the overall variation of entities in $[[q]]$​​​​​​. Finally, we calculate the Spearman’s rank correlation and Pearson’s correlation between $\delta_q$​​​​​​ and the learned apertures of $q$​​​​​​​​​​.
> > >
> > >      - |                             | 1p    | 2p    | 3p    | 2i    | 3i    | pi    | ip    | 2in   | 3in   | inp   | pin   | pni   |
> > >        | --------------------------- | ----- | ----- | ----- | ----- | ----- | ----- | ----- | ----- | ----- | ----- | ----- | ----- |
> > >        | Spearman’s rank correlation | 0.276 | 0.269 | 0.221 | 0.576 | 0.428 | 0.443 | 0.266 | 0.409 | 0.424 | 0.101 | 0.155 | 0.388 |
> > >        | Pearson’s correlation       | 0.404 | 0.278 | 0.248 | 0.537 | 0.489 | 0.420 | 0.292 | 0.332 | 0.378 | 0.086 | 0.112 | 0.302 |
> > >
> > >
> > >
> > > 3. **Additional experiments that compare the neural operations with the geometric set operations**
> > >
> > >    - To make the learning process tractable and the model scalable, we consider the FOL queries in their Disjunctive Normal Form (DNF), which represents FOL queries as a disjunction of conjunctions. Therefore, the union operation can only appear in the last step of a query, and we can hardly turn to the union operation to compute other set operations (e.g., intersection). Moreover, as noted in [Appendix C](https://proceedings.neurips.cc/paper/2020/file/e43739bba7cdb577e9e3e4e42447f5a5-Supplemental.pdf) of BetaE, “queries with negation are only realistic if we take negation with an intersection together”. Taking $\neg A\lor \neg B$​ for an example, a query in this form will look like “List all the entities that are not European countries and all the entities that are not Asian actors”, then both “smartphone” and “cat” will be the answers, which is unrealistic.
> > >
> > >    - Therefore, instead of comparing $\neg(A\land B)$​​​​​​​​​​ with $\neg A\lor \neg B$​​​​​​​​​​, we consider a very similar but more realistic pair $\neg(\neg A\land \neg B)$​​​​​​​​​​ and $A\lor B$​​​​​​​​​​ (just replace $\neg A$​​​​​​​​​​/$\neg B$​​​​​​​​​​ with $A$​​​​​​​​​​/$B$​​​​​​​​​​​), which corresponds to modeling union using De Morgan’s law and the DNF technique. In our Appendix C.2, we have evaluated the performance of these two kinds of modeling. As described in Lines 57-60 in Appendix C.2, ConE using De Morgan’s law performs worse than ConE using DNF. The results are reasonable and expectable. The DNF technique models the exact set union. If we use the complement to handle queries with unions, their representations will always be sector-cones. However, not all queries with union can be well represented by sector-cones (see Figure 3c in the main text).  The results also suggest that we should be careful about the composition orders of logical operations when using a query embedding model, even though they may be logically equivalent.
> > >
> > >    - We conduct experiments to compare the learned embeddings for $\neg (\neg A\land \neg B)$​​ and $A\lor B$​​. We randomly generate $8000$​​ pairs of sector-cones $(A_i, B_i)$​​ and use a ConE model trained on FB15k to generate embeddings for $\neg (\neg A_i\cap \neg B_i)$​​ and $A_i\cup B_i$​​ . Then, to measure the overlap between $\neg (\neg A_i\cap \neg B_i)$​​ and $A_i\cup B_i$​​, we calculate the ratio $r_i=|(\neg (\neg A_i\cap\neg B_i))\cap(A_i\cup B_i) |/|(\neg (\neg A_i\cap\neg B_i))\cup(A_i\cup  B_i)|$​​, and obtain an average ratio of $r=0.4618$​​. The results show a relatively high discrepancy between $\neg (\neg A_i\cap \neg B_i)$​​ and $A_i\cup B_i$​​​, which again validates the results that ConE with DNF technique can outperform ConE with De Morgan’s law.
> > >
> > >
> > >
> > > 4. **The case that the intersection of cones is not a sector-cone**
> > >
> > >    - To find how often the intersection of cones is not a sector-cone, we first randomly generate $8000$​​ pairs of cones $(C_i, D_i)$​​  using queries in the test set of FB15k. At present, the intersection of these cones is not necessarily sector-cones. By calculating their exact set intersections, we find that about $81.8\\%$​​ of the $8000$​​ intersection cones are still sector-cone, accounting for a large proportion.
> > >    - To eliminate the bias in the previous experiments, we calculate the ratio $r_i=|I(C_i, D_i)\cap (C_i\cap D_i)|/|I(C_i, D_i)\cup (C_i\cap D_i)|$​​ over the new $8000$​​ pairs of cones and obtain a new average ratio of $r=0.6134$​​. Note that all the experiments are conducted on the test set, which means that we did not explicitly train our model on these queries. The relatively high overlap ratio $r$ demonstrates that the learned intersection is a good approximation of the real set intersection.
> > >
> > >
> > >
> > > 5. **These departures between “softer” models and geometric constraints should be acknowledged "up-front" and subsequently analyzed to verify the model is behaving in alignment with geometric motivation**
> > >
> > >    - Thanks for the valuable suggestion. We will improve our manuscript based on your suggestions accordingly.

---

> > > > ### Comment · Reviewer_jq2v · 2021-09-02
> > > > **Additional analysis of neural vs. geometric operations as well as logical discrepancies of the model make this a clear accept!**
> > > >
> > > > Thank you so much for the detailed analysis. These additional experiments have completely addressed my concerns. Please do include them in the final version of the paper, which I am very much looking forward to reading, as I believe they add great value and understanding to the model as a whole!
> > > >
> > > > I am increasing my overall rating to an 8.

---

> > > > > ### Author Response · Authors · 2021-09-03
> > > > > **Thanks for your support**
> > > > >
> > > > > Dear Reviewer jq2v,
> > > > >
> > > > > We deeply appreciate your great effort in reading this paper and providing so many insightful comments, which significantly strengthen the quality of this submission. We will include all the results you mentioned in our final version. Thank you again for your kind support!
> > > > >
> > > > >
> > > > >
> > > > > All the best,
> > > > >
> > > > > Authors

---

### Official Review · Reviewer_NFbk · 2021-07-19

**Rating:** 6
**Confidence:** 5

**Summary:**

The paper proposes a new embedding model for multi-hop reasoning over knowledge graphs. The idea is to embed a complex query as n arc segments each on a 1-sphere. The paper designs several neural logic operators including the intersection, projection and complement operators. For union, the model uses the DNF technique to reorder the operators so that the union operators always appear in the last step. The authors evaluate the model on standard benchmarks and achieve better results than prior methods.

Overall the paper is clean and easy to follow. A general question I have is that the current modeling is to embed the query into several independent arc segments. Have you tried to embed the query into the surface (sphere cap) of a n-sphere?


**Limitations And Societal Impact:**

Please check the questions in the section above. I do not see any potential negative societal impact.

**Main Review:**

Here are some additional questions.
1. line 46-47, it seems a little inaccurate. The beta QE model uses the KL divergence to determine whether an entity is an answer to the query. Can you clarify this point?
2. For the SemanticAverage operation, what if the weighted sum of the points is a whole-zero vector? E.g., what is the intersection between $\theta$ and $\theta+\pi$?
3. Can you also report the performance of converting union queries into queries with only intersection and complement operations? How does it compare with the DNF modeling?


**Time Spent Reviewing:**

4

---

> ### Author Response · Authors · 2021-08-10
> **Response to Reviewer NFbk**
>
> We thank the reviewer for the insightful comments. We address the concerns in detail as follows.
>
>
>
> 1. **Embed the query into the surface (sphere cap) of an n-sphere**
>
>    - Thanks for the suggestion. We have not tried this idea and will try it in future work.
>
>
>
> 2. **Lines 46-47: BetaE and the KL divergence**
>
>    - BetaE uses the KL divergence to measure the possibility that an entity is an answer to a query. It can tell us an entity is more likely to be an answer than another entity, but has difficulties in determining whether a given entity is an answer. However, using geometry-based models such as ConE and Query2Box, we can determine an entity as an answer to a given query if the cones/boxes represented the entity are inside the cones/boxes represented the query.
>
>
>
> 3. **SemanticAverage operation**
>
>    - As noted in our paper, we do not need to model all the possible set operations. Specifically, intersecting two entity sets with quite different semantics (e.g., "List the intersection of fruits and sports") makes no sense in real-world applications. Thus, the intersection between $\theta$ and $\theta+\pi$ can hardly happen in real-world applications.
>    - Nonetheless, we can still compute the intersection according to the formula of SemanticAverage. If the learned attention weight vector is not $[0.5, 0.5]$, then the intersection between $\theta$ and $\theta+\pi$ will be an angle between $(\theta, \theta+\pi)$. When the attention weight vector is $[0.5, 0.5]$, then according to the formula in Line 209, the intersection will be $\pi/2$. We have also evaluated the implementation that sets the intersection to $\theta+\pi/2$ when the weight vector is exactly $[0.5, 0.5]$. However, it makes no difference for the final performance as the case rarely happens.
>
>
>
> 4. **The performance of converting union queries into queries with only intersection and complement operations**
>
>    - We report the MRR results of converting union queries into queries with only intersection and complement operations in the following table. The results of BetaE are taken from their original paper [1]. 2u-N/up-N indicates that the disjunction is implemented using the DNF technique, while 2u-M/up-M indicates the implementation using De Morgan's law, i.e., with only intersection and complement operations. The results of ConE on 2u-M/up-M are competitive compared with those of 2u-N/up-N, which all outperform BetaE.  We can also see that ConE using De Morgan's law perform worse than ConE using DNF. The results are reasonable and expectable. If we use the complement to handle queries with unions, their representations will always be sector-cones. However, not all such queries can be well represented by sector-cones (see Figure 3c in the main text). The results can also be found in Appendix C.2.
>
>    - | Dataset | Model | 2u-N     | 2u-M     | up-N     | up-M     |
>      | ------- | ----- | -------- | -------- | -------- | -------- |
>      | FB15k   | BetaE | 40.1     | 25.0     | 25.2     | 25.4     |
>      |         | ConE  | **55.7** | **37.7** | **31.4** | **29.8** |
>      | FB237   | BetaE | 12.4     | 11.1     | 9.7      | 9.7      |
>      |         | ConE  | **14.5** | **13.4** | **10.8** | **9.9**  |
>      | NELL    | BetaE | 12.2     | 11.0     | 8.5      | 8.6      |
>      |         | ConE  | **15.3** | **14.8** | **11.3** | **10.8** |
>
>
>
> [1] Ren, H. and Leskovec, J. Beta embeddings for multi-hop logical reasoning in knowledge graphs. In Neural Information Processing Systems, 2020.

---

> > ### Comment · Reviewer_NFbk · 2021-08-22
> > **Thank you**
> >
> > Thank you for the response and the new results.
> >
> > > However, using geometry-based models we can determine an entity as an answer to a given query if the cones/boxes represented the entity are inside the cones/boxes represented the query.
> >
> > This is an interesting point, if you have saved the checkpoint, can you evaluate the precision/recall of the validation/test queries?

---

> > > ### Author Response · Authors · 2021-08-23
> > > **Thanks for your reply**
> > >
> > > We appreciate your time in reading our response.
> > >
> > > > This is an interesting point, if you have saved the checkpoint, can you evaluate the precision/recall of the validation/test queries?
> > >
> > > The following tables show the precision/recall of the validation/test queries. Note that Query2Box does not apply to queries with negation, so we do not include the corresponding results. The results demonstrate that, using geometry-based models, we can determine whether an entity is an answer to a query by the inclusion relation between entity embeddings and query embeddings. Moreover, ConE outperforms Query2Box on the queries without negation, which is consistent with the results in Table 1 in the main text.
> > >
> > > In this experiment, we use trained models ConE/Query2Box with embedding dimensions $d=800$. That is, each entity and query is represented by a Cartesian product of $800$ cones/boxes. Given an entity embedding $\textbf{v}$ and a query embedding $\textbf{V}_q$, if a majority (we use a threshold of $500$ in the experiments) of the $800$ cones/boxes of $\textbf{v}$ are inside the cones/boxes of $\textbf{V}_q$, we regard the entity $v$ as an answer to the query $q$.  Given a query in the validation/test set, we see its answer entities as positive samples and all the other entities in the knowledge graph as negative samples. Then, we get the precision/recall results in the following tables.
> > >
> > >
> > >
> > > - **Validation Queries**
> > >
> > >   - **Queries without Negation**
> > >
> > >     |               | FB15k       | FB237       | NELL        |
> > >     | ------------- | ----------- | ----------- | ----------- |
> > >     | **Query2Box** | 0.490/0.532 | 0.483/0.533 | 0.458/0.581 |
> > >     | **ConE**      | 0.580/0.678 | 0.519/0.645 | 0.583/0.696 |
> > >
> > >   - **Queries with Negation**
> > >
> > >     |          | **FB15k**   | **FB237**   | **NELL**    |
> > >     | -------- | ----------- | ----------- | ----------- |
> > >     | **ConE** | 0.455/0.648 | 0.545/0.636 | 0.513/0.693 |
> > >
> > >
> > >
> > > - **Test Queries**
> > >   - **Queries without Negation**
> > >
> > >     |               | FB15k       | FB237       | NELL        |
> > >     | ------------- | ----------- | ----------- | ----------- |
> > >     | **Query2Box** | 0.502/0.527 | 0.489/0.506 | 0.466/0.562 |
> > >     | **ConE**      | 0.610/0.670 | 0.536/0.670 | 0.604/0.645 |
> > >
> > >   - **Queries with Negation**
> > >
> > >     |          | **FB15k** | **FB237** | **NELL** |
> > >     | -------- | --------- | --------- | -------- |
> > >     | **ConE** |0.510/0.656|0.560/0.608|0.524/0.627|

---

### Official Review · Reviewer_65PB · 2021-07-19

**Rating:** 7
**Confidence:** 4

**Summary:**

A recent line of knowledge graph (KG) query systems "compiles" the query into an embedded representation, which is interpreted as a geometric region and then used to find response entities whose embedding representations are contained within the region.  Query2box is an early example, where the regions are multidimensional boxes or rectangles.  To be useful, the regions have to be simple, closed under query operators, and capable of being characterized using a few parameters.  Union (disjunction) and negation have therefore created problem for boxes.  This paper replaces boxes with a Cartesian space of cones in 2d.  By restricting the family of permitted cones and disjunction to the last step of any query, the authors argue that their cone family is closed under query operators.  Experiments with benchmarks that have a good diversity of query graph structures show promising results.


**Limitations And Societal Impact:**

Societal impact: N/A

Limitations: Please provide clarification for the questions raised in the main review field.

To a first order, the appeal of this paper is in improved accuracy, not a new modeling or representation insight.


**Main Review:**

Motivation and novelty of problem statement:

The problem statement was established in Query2box.
It is well-motivated, although the claimed "exponential" complexity of discrete graph alignments vs. efficiency of query embedding methods is somewhat debatable.  Plenty of SPARQL implementations do fine on these query classes by clever indexing and pruning.

Moreover, even from the ML perspective, generating a region of embedding space with various assumptions about the compactness of and distances in the region seems more fraught with difficulties than testing a candidate entity against a query, in a discriminative manner.  This also depends on the fanout of a query operator.  See further discussion on table 1 later.  However, the current authors do not really own that debate.


Clarity of presentation:

The paper is quite well written. However, there is one high level issue.  It could well be a limitation in my understanding.

To discuss it, we first need a mild definition.  Let's say a cone in 2d is simple if it has only one contiguous region, i.e., is defined by exactly two rays from the origin.  A simple cone may be convex (if the aperture is at most $\pi$) or non-convex.  A convex cone is always simple.  The complement of a simple cone is also simple.

There is a 1-1 mapping between the Cartesian space of $K$ simple cones and $K$-dimensional boxes.  So, in principle, ConE goes beyond query2box only because they allow a larger family of cones.

The authors define "sector-cones" (definition 3).  A sector-cone or its closure-complement is convex.  If a sector-cone is convex, it is simple.  Otherwise its complement is simple and therefore so is it.  So it seems a sector-cone is always simple.

Now even if there is a bijection between $K$ simple cones and $K$-dimensional boxes, non-convex loss optimization can behave  differently, but it seems to be important to understand the core reason behind the superiority of ConE over Query2box.

Quality and depth of technical proposal/
Detailed comments:

L81 Careful about terminology here: relation instance from $\mathcal{E}$ vs. relation type from $\mathcal{R}$.

L85 triples -> triple

L124 near Defn 3 --- give more examples to clarify the expressiveness boundaries of sector-cones, particularly concerning the discussion about simple cones above.

L150 "should have similar embeddings" --- this is actually quite questionable, and there is some anecdotal evidence (sorry, word of mouth) that even entity sets belonging to what look like coherent types to humans do not occupy tight or compact regions in typical embedding spaces.

L159 remove the two occurrences of "embeddings"

L190 After staying with the paper reasonably well up to this point, and noting the visual definition of intersection in Fig 2(b), SemanticAverage and CardMin come as surprises.  Why should we define the intersection axis as any kind of average?  Aren't the bounding rays of the intersection cone defined in terms of a bunch of min and max operators?

L213 ... and if you are using DeepSets to define the intersection aperture, small wonder about the numbers in Table 3; section 5.3.  I was hoping this would arise naturally out of ConE magic.

L223 "is also a cone" ... but is it a sector-cone?

Table 1 is intuitive.  The trend 1p > 2p > 3p is generally sustained from prior work.  3i gets a large boost, but that is the case even for Q2B --- basically, chaining out is progressively harder with chain length, but intersections are easy (which has been the experience with indexed/combinatorial graph traversal approaches as well).

On negation queries the gains are modest, again raising the question about fundamental model capacity over boxes.


**Time Spent Reviewing:**

4-5 hours

---

> ### Author Response · Authors · 2021-08-10
> **Response to Reviewer 65PB**
>
> We thank the reviewer for the insightful comments. We address the concerns in detail as follows.
>
>
>
> 1. **Motivation of problem statement**
>    - The exponential time complexity of traversing graphs---which does not account for the speedup techniques such as clever indexing and pruning---has been widely acknowledged by researchers in the query embedding community [1,2,3]. We will make it clearer in the revised manuscript.
>    - An important advantage of embedding-based methods is that they can deal with incomplete knowledge graphs, which are common in real-world applications. When some links in knowledge graphs are missing, directly testing a candidate entity against a query may fail.
>
>
>
> 2. **Clarification of presentation**
>    - We note that even if there is a 1-1 mapping between two embedding spaces, the properties of a space and the operators defined in the space can make a big difference for embedding-based models. For example, in the knowledge graph embedding area, RotatE [1]/ComplEx [2] that are in the complex space significantly outperform TransE [3]/DistMult [4] that are in the real space, while there is a 1-1 mapping between the complex space and real space.
>    - The embedding space and the operators are two key parts of a query embedding model. Therefore, we introduce the reason behind the superiority of ConE over Query2Box in these two aspects.
>    - Embedding Space
>
>      - Cones can naturally represent a finite universal set and its subset, while Query2Box cannot. The universal set in a knowledge graph corresponds to the set consisting of all the entities, which is finite. As the apertures of cones are bounded (between $0$ and $2\pi$), we can use the cones with apertures $2\pi$ to represent the universal set and find cones with proper apertures to represent any subsets of the universal set. However, since the offsets of boxes in Query2Box are unbounded, how to find boxes to represent the universal set is unclear. It is worth noting that we cannot constrain the offsets of boxes in Query2Box to be bounded, since the composition of its projection operator can generate boxes with arbitrarily large offsets.
>      - The axes of cones are periodic while the centers of boxes are not. It is an important property to model symmetric relations. We will discuss it in detail in the next part.
>
>    - Operators
>
>      - The operations in the query embedding area usually contain projection, intersection, union, and complement.  The superiority of ConE over Query2Box mainly comes from the projection and complement operator.
>
>      - The projection operator of ConE is more expressive than that of Query2Box. We introduce the superiority in the following two aspects.
>
>        - The projection operator of ConE can generate cones with larger or smaller apertures depending on the relation. However, the projection operator of Query2Box always generates a larger box with a translated center, no matter what the relation is.  In fact, not all the relation projections should result in larger boxes. For example, if an entity set contains all the countries in the world, and the relation is *contain_cities*, the set of adjacent entities will be larger. If the given entity set contains all cities in the world and the relation is *locate_in_country*, the set of adjacent entities will be smaller. Therefore, the projection operator of ConE is more expressive than that of Query2Box. An expressive projection operator can improve the performance on all the queries as projection appears in all query structures.
>        - The projection operator of ConE can well deal with symmetric relations, while the translation-based projection operator of Query2Box cannot. Suppose that $r$ is a symmetric relation. That is, if $r(h,t)$ is true, then $r(t,h)$ will also be true (e.g., *married_with*).  As a corollary, $r^{2n}(h,h)$ should be true for all positive integer $n$.  Using Query2Box, the centers and offsets of the relation $r$ should all be zero. Otherwise, if the offsets are larger than zero, the projected boxes will contain more and more irrelevant entities as $n$ increases. Given that the offsets are zero, the centers of $r$ should also be zero as the centers of $h$ plus $2n$ times centers of $r$ is still the centers of $h$. Therefore, Query2Box has trouble in modeling symmetric relations as it will represent all symmetric relations as zero vectors. For ConE, we can model the symmetric relations by learning a neural operator that rotates some axes by an angle $\pi$ and keeps the apertures unchanged.
>
>      - Since the complements of boxes are no longer boxes, it is still unclear how to use boxes to model the complement operation.
>
>
>
> 3. **L150 "should have similar embeddings"**
>
>    - We expect the learned embeddings of ConE to satisfy the statement, rather than claim that all typical embedding spaces satisfy it.
>
>
>
> 4. **Why should we define the intersection axis as a kind of average**
>
>    - Approximating the intersection operation in a neural way is widely used in query embeddings models [1,2]. As the authors of Query2Box noted in their response to the reviewers (see "Re: Official Blind Review #2 (1/2)" in the OpenReview page of Query2Box), "richer learnable parameterization of the intersection operator is more expressive and also robust to noise in the knowledge graphs".
>
>
>
> 5. **L223: is it a sector-cone**
>
>    - It is a cone (Definition 1) but not a sector-cone. We use sector-cones to represent conjunctive queries, while the disjunction of conjunctive queries is not conjunctive.
>
>
>
> 6. **On negation queries the gains are modest**
>
>    - The relative performance gain of ConE is up to 25.4% against BetaE, which is a significant improvement. The model capacity of ConE is higher than Query2Box as the latter one cannot deal with negation queries.
>
>
>
> 7. **The appeal of this paper**
>
>    - The appeal of our paper comes from both the novel modeling using cones and the significantly improved performance. As noted by Reviewer jq2v, "the approach of using this form of cones is significantly novel". To the best of our knowledge, this is the first geometry-based query embedding model that can handle all the FOL operations, including conjunction, disjunction, and negation.
>
>
>
> 8. **Other suggestions**
>
>    - Thanks for the suggestions. We will improve our paper accordingly.
>
>
>
> [1] Sun, Z., Deng, Z.-H., Nie, J.-Y., and Tang, J. Rotate: Knowledge graph embedding by relational rotation in complex space. ICLR, 2019.
>
> [2] Trouillon, T., Welbl, J., Riedel, S., Gaussier, E., and Bouchard, G. Complex embeddings for simple link prediction. ICML, 2016.
>
> [3] Bordes, A., Usunier, N., Garcia-Duran, A., Weston, J., and Yakhnenko, O. Translating embeddings for modeling multi-relational data. NIPS, 2013.
>
> [4] Yang, B., Yih, W., He, X., Gao, J., Deng, L. Embedding Entities and Relations for Learning and Inference in Knowledge Bases. ICLR, 2015.

---

> > ### Comment · Reviewer_65PB · 2021-08-27
> > **Many thanks for your detailed response!  A few counter-comments.**
> >
> > Motivation: the second point (incomplete knowledge graphs) is the real justification.  We are not yet at a stage with neural graph search that we can disregard indexing and pruning work in combinatorial graph search algorithms.
> >
> > "RotatE [1]/ComplEx [2] that are in the complex space significantly outperform TransE [3]/DistMult [4] that are in the real space, while there is a 1-1 mapping between the complex space and real space."
> >
> > Your sentences above make the situation more murky, unfortunately.  It is not just a matter of mappings between the spaces per se, but the nature of the triple scoring function.  We know that RotatE and ComplEx can deal with asymmetric relations, whereas DistMult cannot.  We know that ComplEx can deal with many-to-many relations, whereas TransE cannot.  So this is not an argument or explanation for why ConE performs better than Query2box.
> >
> > I request a further discussion of some properties of sector cones and their implications.
> >
> > "since the offsets of boxes in Query2Box are unbounded, how to find boxes to represent the universal set is unclear."
> >
> > One need not formulate Query2Box to have unbounded offsets, even if that's conventional.  (The "boxes" need not even lie in a standard normed Euclidean space, but let us not get into that.)  One could, for example, pass the upper and lower bounds of the boxes through a sigmoid to contain all boxes in $[0,1]^D$.  Query2Beta uses the same range, apparently without deleterious effects.
> >
> > A convex cone (at least in 2d) cannot have two disconnected regions (connected only at the origin via closure).  Such a cone is not simple (see my initial review).  The complement of a non-simple cone is also not simple.  So it seems to me that a sector-cone must be simple, even if it is not convex.  That is why you can always assign it a symmetric axis.
> >
> > The union of sector-cones is not necessarily simple.  The complement of a sector cone is always simple.  They are still cones (L126).
> >
> > Also, I am sorry that it is not yet clear to me why the wrapping around of cones with a period of $2\pi$ is so important.  Does a Query2X system really need to take advantage of such a capability?
> >
> > If we discount wrap-around, the space of simple cones is topologically equivalent to boxes in $[0,1]^D$, because you regard the $D$ dimensions of the cone embedding as independent; i.e., there are no cones sitting in $\mathbb{R}^D$, only $\mathbb{R}^2$.
> >
> > Overall, ConE is clearly doing something useful, but I do not find the "engineering minimality" I am looking for.  Do we have to think of the setup with cones, are sector-cones the fundamental trick that improved numbers, or is the crux of the improvements somewhere else?  I generally like the paper, but clearer thinking along the above lines would make it even better.

---

> > > ### Author Response · Authors · 2021-08-29
> > > **Thanks for your further comments**
> > >
> > > We appreciate your time and efforts for the further discussion. We have carefully read your comments. Below is our response and we hope it can properly address your concerns.
> > >
> > >
> > >
> > > 1. **RotatE and ComplEx can deal with asymmetric relations, whereas DistMult cannot. ComplEx can deal with many-to-many relations, whereas TransE cannot.**
> > >
> > >    - We compare RotatE with TransE, and ComplEx with DistMult. Both RotatE and TransE can deal with asymmetric relations, and both ComplEx and DistMult can deal with many-to-many relations. Although there is a 1-1 mapping between the spaces of RotatE and TransE and a 1-1 mapping between the spaces of ComplEx and DistMult, RotatE and ComplEx significantly outperform TransE and DistMult, respectively.
> > >
> > >
> > >
> > > 2. **It is not just a matter of mappings between the spaces per se, but the nature of the triple scoring function. So this is not an argument or explanation for why ConE performs better than Query2box.**
> > >
> > >    - Our point is that even if there exists a 1-1 mapping between two spaces, different spaces allow us to define different scoring functions on them, which have a significant impact on models' performance. For example, ComplEx outperforms DistMult as its scoring function uses the complex conjugate of entity embeddings to handle asymmetric relations. Since we can only perform complex conjugation in a complex space, the nature of this triple scoring function is strongly related to the embedding space.
> > >    - For query embedding models, even if it is possible to construct a 1-1 mapping between a cone space and a box space, different spaces allow us to define different operators and distance functions. For example, we can define the complement operator for queries with negation in the cone space, while we cannot do it in the box space. Moreover, in the cone space, we can naturally represent finite universal sets and define a projection operator that can deal well with symmetric relations. That is why ConE outperforms Query2Box.
> > >
> > >
> > >
> > > 3. **One need not formulate Query2Box to have unbounded offsets. One could pass the upper and lower bounds of the boxes through a sigmoid to contain all boxes in $[0,1]^D$.**
> > >
> > >    - Thanks for sharing this interesting point. To the best of our knowledge, we have not seen any existing work on query embedding using bounded boxes. We will try to explore and leave it to a future work.
> > >    - It is possible that, even if we pass the upper and lower bounds of the boxes through a sigmoid to make the offsets bounded and contain all boxes in $[0,1]^D$, the embedding parameters before being passed through the sigmoid are unbounded. Thus, we still have difficulty in finding proper box parameters to represent the finite universal set, i.e., the set containing all the entities in a knowledge graph.
> > >
> > >
> > >
> > > 4. **Query2Beta uses the same range $[0,1]^D$​, apparently without deleterious effects.**
> > >
> > >     - The embedding parameter range of BetaE (Query2Beta in the review) is also unbounded. BetaE relies on the density of a Beta distribution $p(x)=\frac{x^{\alpha-1}(1-x)^{\beta-1}}{B(\alpha,\beta)}$​, where $x\in[0,1]$​ and $B(\cdot)$​ denotes the beta function. BetaE uses the parameters $(\alpha, \beta)$​ to represent entities and queries, both of which take values in $(0,+\infty)$​. That is to say, the embedding parameter range of BetaE is $(0,+\infty)^D$​​​​.
> > >
> > >
> > >
> > > 5. **Why the wrapping around of cones with a period of $2\pi$​​​​​ is important.**
> > >
> > >     - As discussed in our previous response, the periodicity of cones is a helpful property to model symmetric relations.  Suppose that $r$​​​ is a symmetric relation. That is, if $r(h,t)$​​​ is true, then $r(t,h)$​​​ will also be true (e.g., *married_with*).  Suppose that the embedding dimension $d=1$​​​, the axis of $h$​​​ is $\theta_h$​​​, the axis of $t$​​​ is $\theta_t$​​​, and the apertures of $h$​​​ and $t$​​​ are $0$​​​. Then, ConE can model the symmetric relation by learning a neural operator that rotates some axes by an angle $\pi$​​​ and keeps the apertures unchanged. That is, ConE can model the relation $r$​​​ between $h$​​​ and $t$​​​ as $r(\theta_h)=\theta_h+\pi=\theta_t$​​​ and $r(\theta_t)=\theta_t+\pi=\theta_h$​​, which is benefited from the periodicity. A similar case can be found in RotatE. RotatE can deal with symmetric relations since the phases in complex spaces are periodic.
> > >     - The periodicity of cones also gives us opportunities to model queries with negation. The "wrapping around" property of cones implies that the closure-complement of a sector-cone is a sector-cone, depending on which we can model queries with negation.
> > >
> > >
> > >
> > > 6. **Clearer thinking along the above lines would make this paper even better.**
> > >
> > >    - Thanks for the valuable suggestion. We will improve our paper accordingly.

---

> > > > ### Comment · Reviewer_65PB · 2021-09-02
> > > > **Thanks for the additional explanations**
> > > >
> > > > I agree with much of what you said in your latest reponse.
> > > > I also read relevant parts of your responses to all other reviewers.
> > > > Based on the interaction, I am raising my score from 6 to 7.
> > > > Although 7 perhaps means "absolute accept", there might be other submissions that  supersede this.  (I.e., I'd like this paper to get in, but won't fight for it.)  But at least I understand the paper a bit better.  Thanks!
> > > >
> > > > About ICML 2021, you wrote "The reason for rejection is that the motivation concept of our paper is a bit confusing."
> > > > The number of rounds of discussion we had shows that this limitation has not yet been perfectly fixed, but the detailed discussion should give you enough ammo to fix this completely --- that would be my request.
> > > >
> > > > A few final clarifications from my end.
> > > >
> > > > Yes, just passing the box limits through a sigmoid may not ease any optimization, in fact it could make it worse.  But whatever you are doing right in cone space might be possible to map back to box space, except for...
> > > >
> > > > ...the wraparound and its effect on complementation.  I feel this is where the crux is. Other distinctions with boxes are less central.
> > > >
> > > > I didn't mean the parameters of Beta were bounded in [0,1], just that the support of Beta is, just like the straw proposal of passing endpoints via sigmoid to get boxes trapped inside the unit cube.
> > > >
> > > > "different spaces allow us to define different scoring functions" and you gave the example of DistMult vs ComplEx.  However, complex conjugacy just happens to be a convenient aid to exposition, it does not really "mean anything" about the complex embedding vectors, which remain opaque anyway.  Note that even with real embeddings,  one might have split them up into sub-vectors and introduced an asymmetric scoring function.  In fact, holographic embeddings used no complex formalism and was algebraically equivalent to ComplEx.

---

> > > > > ### Author Response · Authors · 2021-09-03
> > > > > **Thanks for your support**
> > > > >
> > > > > Dear Reviewer 65PB,
> > > > >
> > > > >
> > > > >
> > > > > Many thanks for your great effort in reviewing our paper and replying to our response. We believe your valuable suggestions will significantly strengthen the quality of this submission. We will try our best to improve our manuscript based on the discussions in the rebuttal period. Thank you again for your kind support!
> > > > >
> > > > >
> > > > >
> > > > > All the best,
> > > > >
> > > > > Authors

---

### Official Review · Reviewer_uQKs · 2021-07-21

**Rating:** 6
**Confidence:** 2

**Summary:**

The paper presents a novel geometric embeddings called cone embeddings for solving query answering. Query answering requires embeddings of queries that are represented using first order logic. This includes conjunction, disjunction, negation but excluding Universal Quantification. Given this as basis, the related work until now has focused on two different types of embeddings (1) Geometric; and (2) probabilistic query based model. The progression from Geometric based embeddings to probabilistic models was primarily due to the fact that geometric embeddings cannot handle negation.

**Main Review:**

The work differentiates itself from the other related works in the geometric query embeddings space by handling negation in first order logic queries for knowledge graphs. They propose cone embeddings (in comparison to Box Embeddings). Cone embeddings can handle negation because complement of a cone is also a cone, whereas this is the primary limitation for box embeddings. The approach also proposes four operators (a) Projection, (b) Intersection, (c) Union, and (d) Complement. The experiments on three different datasets and 9 different patterns of queries shows that Cone embeddings outperform the other approaches including BetaE, Query2Box, and GQE. This also includes its performance gain in predicting the cardinality of the answers.

The paper is written well and the overall evaluation shows significant performance gain in comparison to other approaches.

My main concerns with the paper are as follows:
1. Comparison to faithful embeddings: The primary drawbacks of query embeddings approach is that they focus on incomplete knowledge bases. In pursuit of this -- they lose significant performance on complete knowledge bases. This is an important feature that is required for all embeddings. This work does not compare itself on the setting and also does not provide any reasons why they did not attempt comparing.
2. Qualitative analysis between Box Embeddings and Cone Embeddings: While Cone Embeddings makes sense for negation, it is unclear why they perform significantly better than Box Embeddings. This is also the case in comparison to BetaE for negation. The authors primarily drive the work as having a geometric embeddings for negation but do not motivate/justify why Cone embeddings performs better than Box Embeddings for this work. Particularly, seems like Box Embeddings can handle most of the conjunctive queries without negation and is probably more easily interpretable than cone embeddings.

Overall -- its hard for me to understand why ConeEmbeddings performs well on all aspects since it is not clearly explained in the paper. Further, it is important to have the time taken for training both ConeEmbeddings, BetA, and BoxEmbeddings. This is primarily because of the effect of multiple operators including the negation in Cone embeddings.

**Time Spent Reviewing:**

5 hrs

---

> ### Author Response · Authors · 2021-08-10
> **Response to Reviewer uQKs**
>
> We thank the reviewer for the insightful comments. We address the concerns in detail as follows.
>
>
>
> 1. **Comparison to faithful embeddings**
>
>    - In our submission, we follow the experimental settings in GQE [1], Q2B [2], and BetaE [3] that train models on incomplete knowledge graphs (i.e., the "generalization" setting). We conduct more experiments on FB15k under the setting that trains models on complete knowledge graphs (i.e., the "entailment" setting) and compare ConE with the state-of-the-art faithful embeddings EmQL [4]. The following table shows the MRR results, where the results of Q2B, EmQL, and EmQL-sketch are taken from [4]. We can see that ConE significantly outperforms Q2B and EmQL-sketch, showing the faithfulness of ConE embeddings.  It is expectable that EmQL outperforms ConE under this setting, since it is designed expressly for modeling faithfulness. However, as shown in Appendix C.4, ConE significantly outperforms EmQL under the "generalization" setting ($0.528$ vs. $0.439$). It is promising that using the sketch technique proposed in [4] to further improve the faithfulness of ConE and we leave it as future work.
>
>    - | Models      | 1p    | 2p    | 3p    | 2i    | 3i    | ip    | pi    | 2u    | up    | AVG   |
>      | ----------- | ----- | ----- | ----- | ----- | ----- | ----- | ----- | ----- | ----- | ----- |
>      | Q2B         | 0.559 | 0.347 | 0.288 | 0.389 | 0.553 | 0.145 | 0.280 | 0.444 | 0.257 | 0.362 |
>      | EmQL        | 0.983 | 0.961 | 0.908 | 0.908 | 0.872 | 0.881 | 0.883 | 0.887 | 0.910 | 0.910 |
>      | EmQL-sketch | 0.819 | 0.448 | 0.368 | 0.564 | 0.580 | 0.420 | 0.466 | 0.385 | 0.383 | 0.493 |
>      | ConE        | 0.766 | 0.579 | 0.494 | 0.568 | 0.659 | 0.314 | 0.448 | 0.636 | 0.412 | 0.542 |
>
> ​
>
> 2. **Qualitative analysis for Cone Embeddings**
>    - **Comparison with Query2Box**
>      - The embedding space and the operators are two key parts of a query embedding model. Therefore, we introduce the superiority of ConE over Query2Box in these two aspects.
>
>      - Embedding Space
>
>        - Cones can naturally represent a finite universal set and its subset, while Query2Box cannot. The universal set in a knowledge graph corresponds to the set consisting of all the entities, which is finite. As the apertures of cones are bounded (between $0$ and $2\pi$), we can use the cones with apertures $2\pi$ to represent the universal set and find cones with proper apertures to represent any subsets of the universal set. However, since the offsets of boxes in Query2Box are unbounded, how to find boxes to represent the universal set is unclear. It is worth noting that we cannot constrain the offsets of boxes in Query2Box to be bounded, since the composition of its projection operator can generate boxes with arbitrarily large offsets.
>        - The axes of cones are periodic while the centers of boxes are not. It is an important property to model symmetric relations. We will discuss it in detail in the next part.
>
>      - Operators
>
>        - The operators in query embedding models usually contain projection, intersection, union, and complement.  The superiority of ConE over Query2Box mainly comes from the projection and complement operator.
>        - The projection operator of ConE can generate cones with larger or smaller apertures depending on the relation. However, the projection operator of Query2Box always generates a larger box with a translated center, no matter what the relation is.  In fact, not all the relation projections should result in larger boxes. For example, if an entity set contains all the countries in the world, and the relation is *contain_cities*, the set of adjacent entities will be larger. If the given entity set contains all cities in the world and the relation is *locate_in_country*, the set of adjacent entities will be smaller. Therefore, the projection operator of ConE is more expressive than that of Query2Box. An expressive projection operator can improve the performance on all the queries as projection appears in all query structures.
>        - The projection operator of ConE can well deal with symmetric relations, while the translation-based projection operator of Query2Box cannot. Suppose that $r$ is a symmetric relation. That is, if $r(h,t)$ is true, then $r(t,h)$ will also be true (e.g., *married_with*).  As a corollary, $r^{2n}(h,h)$ should be true for all positive integer $n$.  Using Query2Box, the centers and offsets of the relation $r$ should all be zero. Otherwise, if the offsets are larger than zero, the projected boxes will contain more and more irrelevant entities as $n$ increases. Given that the offsets are zero, the centers of $r$ should also be zero as the centers of $h$ plus $2n$ times centers of $r$ is still the centers of $h$. Therefore, Query2Box has trouble in modeling symmetric relations as it will represent all symmetric relations as zero vectors. For ConE, we can model the symmetric relations by learning a neural operator that rotates some axes by an angle $\pi$ and keeps the apertures unchanged.
>
>        - Since the complements of boxes are no longer boxes, it is still unclear how to use boxes to model the complement operation.
>    - **Comparison with BetaE for negation**
>      - ConE outperforms BetaE on queries with negation as ConE can represent the universal set in a knowledge graph. By definition, the computation of set complement is closely related to the universal set. Accordingly, to better model queries with negation, a query embedding model should be capable of representing the universal set. For ConE, we can use cones with apertures $2\pi$ to represent the universal set. However, for BetaE, it is unclear how to use a Beta distribution to achieve the goal.
>
>
>
> 3. **Time taken for training**
>
>    - To evaluate the training speed of ConE and all the baselines, we report the average time spent to run 100 training steps. We run all the models with the same number of embedding parameters using a single RTX 3090 GPU card. The following table demonstrates that the simplest model GQE is the most time-efficient. The training speed of ConE is close to that of Query2Box (Q2B) and faster than BetaE.
>
>    - | Models | Average Time Spent/100 Training Steps |
>      | ------ | ------------------------------------- |
>      | GQE    | 15s                                   |
>      | Q2B    | 17s                                   |
>      | BetaE  | 28s                                   |
>      | ConE   | 21s                                   |
>
>
>
> [1] Hamilton, W. L., Bajaj, P., Zitnik, M., Jurafsky, D., and Leskovec, J. Embedding logical queries on knowledge graphs. NeurIPS, 2018.
>
> [2] Ren, H., Hu, W., and Leskovec, J. Query2box: Reasoning over knowledge graphs in vector space using box embeddings. ICLR, 2020.
>
> [3] Ren, H. and Leskovec, J. Beta embeddings for multi-hop logical reasoning in knowledge graphs. NeurIPS, 2020.
>
> [4] Sun, H., Arnold, A. O., Bedrax-Weiss, T., Pereira, F., and Cohen, W. W. Faithful embeddings for knowledge base queries. NeurIPS, 2020.

---

### Official Review · Reviewer_Zjbx · 2021-07-23

**Rating:** 6
**Confidence:** 4

**Summary:**

The paper proposes a new framework for embedding first-order logic (FOL) queries over knowledge bases using region-based embeddings. The embedding space is a Cartesian product of 2D cones. Entities are embedded in the same space with 0-aperture cones (ray starting at origin).

- Authors provide the relevant foundation to cones in 2D space, the final Cartesian product space, and suggest parameterizations of projection, intersection, union, and complement operations.
- The choice of using cones facilitates complement operation.
- Authors demonstrate empirical improvements over past work in FOL query embedding.
- While the proposed cone-based region embeddings allow for clear descriptions for intersection, union, and complement, the implementation does not strictly enforce these geometric concepts.

**Limitations And Societal Impact:**

### Limitations

(1)

The set of sector cones (used to represent the query) is not closed under union and intersection.

(2)

The proposed parameterization of the intersection operation does not correspond exactly to set intersection.

As a concrete example, consider a space of a single 2D cone (i.e. *d*=1). The intersection of two sector cones with $C_1$ = ($\theta_{ax} = 0$, $\theta_{ap} = \pi / 2$) and $C_2$ = ($\theta_{ax} = \pi / 4$, $\theta_{ap} = \pi / 2$), have an intersection of $C_I$ = ($\theta_{ax} = \pi / 8$, $\theta_{ap} = \pi / 4$). This is not guaranteed by the parameterization i.e. an entity can belong to $C_1$ and $C_2$ but not to $I(C_1, C_2)$ as defined in sec 4.2.

(3)

The definition of an entity belonging to the answer set is not clear. This is especially confusing when using $d > 1$. The use of both *inside* and *outside* distance in the scoring seems to convey that an entity can belong to the answer set as long as it intersects any of the *d* cones in the query embedding. In this case, an entity can have some overlapping components with both the query and the complement of the query.

Alternatively, we can define that an entity belongs to the answer set only if it intersects all of the *d* cones in the query embedding. Under this definition, the proposed complement operation is too restrictive since an entity with just one mismatched cone dimension should belong to the complement.

- - -

The geometric embeddings help build intuitions/mental pictures when *d*=1 but the geometry isn't strictly imposed always. It would be useful to explicitly discuss this along with barriers to implementing fully geometric operations.

### Societal Impact

The work does not pose issues in terms of negative societal impacts besides concerns noted in the supplementary materials.

**Main Review:**

### Originality

- The work builds on past work on FOL query embedding and suggests new parameterization using cones. Set complements of cones are also cones and lie in the same embedding space. This was a limitation of past work that relied on geometrical embeddings (e.g. complement of box embeddings is no longer a box). This choice allows the representation to model all FOL operations.
- It has appropriately cited past related work and placed itself with respect to it.

### Quality

- The representation capacity of the proposed method is demonstrated with empirical results. The baselines are fairly compared. The ability to tackle set complement is also compared with relevant baselines.
- **(1) Minor clarification**: ConE embeddings are trained with *d*=800. Is this space the Cartesian product of 800 cones (so a total of 1600 parameters) or 400 cones (800 parameters)? If is it the former, then shouldn't the baselines also have 1600 dimensional embeddings?
- **(2) Strong claim**: Line 297 claims that cone embeddings are better geometric representations than box embeddings. I believe this is a very strong claim since it only compares against specific parameterizations of box embeddings. In particular, the notion of intersection and union of box embeddings is still an active area of research and there isn't a universally accepted parameterization.

### Clarity

- The paper provides the necessary mathematical background to understand the proposed approach of conical embeddings.
- Hyperparameters are reported in sufficient detail to recreate the experimental results.

### Significance

- The work improves the state of the art in a demonstrable way.
- There is scope to experiment with other parameterizations of the same underlying conical embeddings space for marginal improvements.
- There is also scope to view the apertures in a probabilistic framework (the fraction of the entire space covered by a cone can be interpreted as likelihood similar to probabilistic box embeddings)

**Time Spent Reviewing:**

4

---

> ### Author Response · Authors · 2021-08-10
> **Response to Reviewer Zjbx**
>
> We thank the reviewer for the insightful comments. We address the concerns in detail as follows.
>
>
>
> 1. **Clarification for the embedding dimensions**
>
>    - We use the Cartesian product of $800$ cones---i.e., a total of $1600$ parameters---in the experiments.
>    - We ran all the baselines, including GQE, Query2Box, and BetaE, with $1600$ parameters. As Query2Box/BetaE represents a box/Beta distribution with two parameters, $1600$ parameters also correspond to $d=800$. However, as pointed out in Lines 283-286, BetaE with $d=800$ performs worse than that with $d=400$. Therefore, we report the performance of BetaE with $d=400$ in the main text and include the results for $d=800$ in Appendix C.1.
>    - In Line 285, the embedding dimension of GQE should be $d=1600$. Thanks for pointing out the typo.
>
>
>
> 2. **Clarification for the claims in Line 297**
>
>    - Box embeddings here refer to the Query2Box model instead of all the possible box embeddings, since Query2Box is the most popular box embeddings for the query embedding community. We will clarify it in our revised manuscript.
>
>
>
> 3. **The set of sector cones (used to represent the query) is not closed under union and intersection.**
>
>    - It is unnecessary that the set of sector-cones is closed under union and intersection. We note that, we aim to represent entities and all the possible queries, instead of all the theoretically possible sets. Moreover, sector-cones are used to represent conjunctive queries instead of all the queries.
>     - The union operation corresponds to the disjunction of conjunctive queries. Since the disjunction of conjunctive queries is no longer conjunctive, we need not represent them with sector-cones. Moreover, as the disjunction only appears in the last step of queries, whether the set of cones is closed under union does not affect the composition of logical operators.
>     - The intersection operation corresponds to the conjunction of conjunctive queries, which is still conjunctive. Although the intersection of sector-cones may not be a sector-cone, the case can hardly happen in practice. We expect entities with similar semantics to locate in a contiguous "region" in the embedding space. The case that the intersection of sector-cones is not a sector-cone corresponds to that the answers to the conjunction of conjunctive queries have dissimilar semantics, which is uncommon in real-world applications.
>
>
>
> 4. **The proposed parameterization of the intersection operation does not correspond exactly to the set intersection.**
>
>    - Approximating the intersection operation in a neural way is widely used in query embeddings models [1,2]. As the authors of Query2Box noted in their response to the reviewers (see "Re: Official Blind Review #2 (1/2)" in the [OpenReview](https://openreview.net/forum?id=BJgr4kSFDS&noteId=SyxzjxvjsH) page of Query2Box), "richer learnable parameterization of the intersection operator is more expressive and also robust to noise in the knowledge graphs". The neural intersection operator can approximate the set intersection via the learning process. For example, if an entity $v$ belongs to $C_1$ and $C_2$ but not to $I(C_1, C_2)$, then both the outside and inside distances between $e$ and $I(C_1, C_2)$ are nonzero. Therefore, by minimizing the distance function, $v$ will tend to be inside $I(C_1, C_2)$.
>    - Empirically, we conduct an experiment to evaluate how well the neural operator approximate the exact set intersection. We randomly generate 8000 pairs of sector-cones $(C_i, D_i)$, where $C_i\cap D_i$ is also a sector-cone. Then, we use a ConE model trained on FB15k to generate embeddings $I(C_i, D_i)$ for the intersection $C_i\cap D_i$. Ideally, the learned cones $I(C_i, D_i)$ should be the same as the real cones $C_i\cap D_i$. We calculate the ratio $r_i=|I(C_i, D_i)\cap (C_i\cap D_i)|/|I(C_i, D_i)\cup (C_i\cap D_i)|$ to measure the overlap between $I(C_i, D_i)$ and $C_i\cap D_i$, and obtain an average ratio of $r=0.7436$. That is to say, the learned intersection $I(C_i, D_i)$ is a good approximation of the real intersection $C_i\cap D_i$.
>
>
>
> 5. **The definition of an entity belonging to the answer set is not clear.**
>
>    - Whether an entity $v$ belongs to the answer set $[[q]]$ is determined by the outside distance $d_o(\textbf{V},\textbf{V}_q)$, where $\textbf{V}$ and $\textbf{V}_q$ are the cone embeddings for the entity $v$ and query $q$, respectively. Ideally, the entity $v$ belongs to the answer set $[[q]]$  when $d_o(\textbf{V},\textbf{V}_q)=0$, i.e., the entity embedding $\textbf{V}$ intersects all of the cones in the query embedding $\textbf{V}_q$. Accordingly, in the ideal case, an entity belongs to the complement if it intersects all of the cones in the negation query embedding. However, we allow some components of $\textbf{V}$ outside the corresponding components of $\textbf{V}_q$ in practice. If $d_o(\textbf{V},\textbf{V}_q)$ is small enough (e.g., smaller than a threshold), we can recognize the entity $v$ as an answer to the query $q$. Moreover, in this way, even if we have two entities that both have mismatched cones, we can say that one entity is more likely to be the answer than the other one by comparing their distances to the query embeddings.
>    - We cannot say an entity with just one mismatched cone dimension belongs to the complement, since the entity does not intersect all of the $d$ cones in the negation query embeddings as well. However, we can compare the possibility that it belongs to the query or its complement by comparing the distances between the entity and queries.
>
>
>
> 6. **Other suggestions**
>
>    - Thanks for the suggestions. We will improve our paper accordingly.
>
>
>
> [1] Ren, H., Hu, W., and Leskovec, J. Query2box: Reasoning over knowledge graphs in vector space using box embeddings. ICLR, 2020.
>
> [2] Ren, H. and Leskovec, J. Beta embeddings for multi-hop logical reasoning in knowledge graphs. NeurIPS, 2020.

---

### Author Response · Authors · 2021-08-22
**Please let us know if you have any feedback and/or additional questions.**

We thank all the reviewers for the insightful comments and constructive suggestions, which are very helpful for us to strengthen this submission.

We have submitted detailed responses to address the concerns raised by all the reviewers. Besides answering the technical questions,

1. we elaborate on the relationship between the proposed operators and the exact set operations, and clarify the definition of an entity belonging to the answer set (Reviewer Zjbx);
2. we compare ConE with faithful embeddings and provide qualitative analysis between ConE and Query2Box (Reviewer uQKs);
3. we clarify the motivation of the problem statement and the presentation of model design (Reviewer 65PB);
4. we report the performance of converting union queries into queries with only intersection and complement operations (Reviewer NFbk);
5. we clarify our motivation of using cones and demonstrate that the designed operators can well approximate the real set operations (Reviewer jq2v).

We would like to know if our responses have properly addressed your concerns. All of your feedback and/or additional comments are warmly welcomed.

---

### Decision · Program_Chairs · 2021-09-27

**Decision:**

Accept (Poster)

**Comment:**

The paper attempts to improve reasoning over knowledge bases. In this regard, the authors introduces a novel query representation using cones. As the set of cones is closed under intersection, union, and complement, it is claimed that such representation can be very expressive. Empirical performance of the proposed method is very good. We thank the reviewers and authors for engaging in an active discussion, which resulted in clearing a lot of the concerns (e.g. mismatch between motivation and method) and a lot of constructive feedback were provided to improve the paper. The authors provided extensive empirical results as part of the discussion, please include them in the final version of the paper as they add great value and understanding to the model as a whole.